# Aerobic exercise reverses aging-induced depth-dependent decline in cerebral microcirculation

**Paul Shin[1]\*, Qi Pian[1], Hidehiro Ishikawa[2], Gen Hamanaka[2], Emiri T Mandeville[2], Shuzhen Guo[2], Buyin Fu[1], Mohammed Alfadhel[1,3], Srinivasa Rao Allu[4,5], Ikbal Şencan-Eğilmez[1,6], Baoqiang Li[1,7], Chongzhao Ran[1], Sergei A Vinogradov[4,5], Cenk Ayata[8,9], Eng Lo[2], Ken Arai[2], Anna Devor[1,10], Sava Sakadžić[1]**

[1]Athinoula A Martinos Center for Biomedical Imaging, Massachusetts General Hospital, Harvard Medical School, Charlestown, United States; [2]Neuroprotection Research Laboratory, Departments of Radiology and Neurology, Massachusetts General Hospital, Harvard Medical School, Charlestown, United States; [3]Department of Bioengineering, Northeastern University, Boston, United States; [4]Department of Biochemistry and Biophysics, University of Pennsylvania, Philadelphia, United States; [5]Department of Chemistry, University of Pennsylvania, Philadelphia, United States; [6]Biophotonics Research Center, Mallinckrodt Institute of Radiology, Department of Radiology, Washington University School of Medicine, St. Louis, United States; [7]Brain Cognition and Brain Disease Institute, Shenzhen Institute of Advanced Technology, Chinese Academy of Sciences, Shenzhen, China; [8]Neurovascular Research Laboratory, Department of Radiology, Massachusetts General Hospital, Harvard Medical School, Charlestown, United States; [9]Stroke Service, Department of Neurology, Massachusetts General Hospital, Harvard Medical School, Charlestown, United States; [10]Department of Biomedical Engineering, Boston University, Boston, United States

**\*For correspondence:**
pshin2@mgh.harvard.edu

**Competing interest:** The authors declare that no competing interests exist.

**Abstract** Aging is a major risk factor for cognitive impairment. Aerobic exercise benefits brain function and may promote cognitive health in older adults. However, underlying biological mechanisms across cerebral gray and white matter are poorly understood. Selective vulnerability of the white matter to small vessel disease and a link between white matter health and cognitive function suggests a potential role for responses in deep cerebral microcirculation. Here, we tested whether aerobic exercise modulates cerebral microcirculatory changes induced by aging. To this end, we carried out a comprehensive quantitative examination of changes in cerebral microvascular physiology in cortical gray and subcortical white matter in mice (3–6 vs. 19–21 months old), and asked whether and how exercise may rescue age-induced deficits. In the sedentary group, aging caused a more severe decline in cerebral microvascular perfusion and oxygenation in deep (infragranular) cortical layers and subcortical white matter compared with superficial (supragranular) cortical layers. Five months of voluntary aerobic exercise partly renormalized microvascular perfusion and oxygenation in aged mice in a depth-dependent manner, and brought these spatial distributions closer to those of young adult sedentary mice. These microcirculatory effects were accompanied by an improvement in cognitive function. Our work demonstrates the selective vulnerability of the deep cortex and subcortical white matter to aging-induced decline in microcirculation, as well as the responsiveness of these regions to aerobic exercise.

### Editor's evaluation
Using convincing approaches with mice, the authors show that aging is associated with a reduction in microvascular perfusion and oxygenation and that voluntary aerobic exercising restored these parameters, especially in the white matter. This work is of broad interest to medical biologists in the field of cerebrovascular diseases.

## Introduction

The cerebral white matter is significantly affected by aging (*Cees De Groot et al., 2000*; *de Leeuw et al., 2001*). Previous work suggests that, in comparison with superficial gray matter, subcortical white matter has greater tissue loss due to aging and is more susceptible to hypoperfusion (*Gunning-Dixon et al., 2009*; *Hase et al., 2019*; *Li et al., 2020*). Indeed, white matter lesions might impair cognitive function more than gray matter lesions (*Reber et al., 2021*), underscoring the importance of understanding the biological mechanisms of aging-related white matter degeneration for targeted interventions.

Cerebral small vessel disease (CSVD) refers to a range of pathological processes affecting small arterioles, capillaries, and venules supplying the white matter and deep structures of the brain. CSVD is a common cause of stroke and an important contributor to age-related cognitive decline and dementia (*Pantoni, 2010*; *Smith and Markus, 2020*). Most CSVD-related strokes affect subcortical white and deep gray matter. In addition, there is increased awareness of the role of CSVD in accelerating the pathogenesis of Alzheimer's disease (AD), with some studies suggesting the possibility that white matter changes are the starting point of AD (*Defrancesco et al., 2014*; *Esiri et al., 1999*; *Radanovic et al., 2013*; *Snowdon et al., 1997*). Despite this growing awareness, our mechanistic understanding of aging-related changes in microcirculation as a function of depth in cortex and underlying white matter is incomplete.

Aerobic exercise is a promising strategy to improve neurocognitive function in aging and reduce the risk of age-related neurological disorders (*Barnes et al., 2003*; *Yaffe et al., 2001*). Most studies have thus far focused on exercise-responsive molecules that could lead to improvement of neural physiology and, thereby, cognitive performance (*De Miguel et al., 2021*; *Islam et al., 2021*; *Valenzuela et al., 2020*; *Wang and Holsinger, 2018*). Yet, how exercise may exert beneficial effects on vascular contributions to brain aging remain to be fully understood. In particular, it is unclear how exercise normalizes cerebral microcirculation, and whether there are differential effect across brain regions, including the gray and white matter.

In this study, we compared young (3–6 months old) versus old (19–21 months old) mice to assess the effects of normal aging, and then asked whether 5 months of voluntary aerobic exercise can alter or rescue brain microcirculation in old 20-month-old mice. Cerebral microvascular perfusion and oxygenation in the cortex and subcortical white matter were quantified with two-photon microscopy (2PM) and optical coherence tomography (OCT) in awake mice.

Our results showed age-related declines in capillary red-blood-cell (RBC) flux and capillary oxygen partial pressure ($pO_2$) in the deep cortical layers and subcortical white matter, while voluntary exercise improved these measures, including the cerebral blood flow (CBF) in cortical ascending venules. Interestingly, the regions that experienced the highest decline were also the ones that benefited the most from the exercise. These microcirculatory effects were accompanied by an improvement in cognitive function. Our results may provide insights into how sedentary aging and aerobic exercise affect cerebral microvascular physiology and particularly emphasize the physiologic importance of effects in the deep cortex and subcortical white matter.

## Results

### Exercise mitigates age-related decline of capillary RBC flux and induces capillary flow homogenization in the subcortical white matter

To examine the impact of aging on cortical and subcortical microcirculation, we examined the spatial distribution of capillary RBC flux across cortical and subcortical regions in aged mice and compared the result with measurements from awake C57BL/6N female mice of younger age (4 months old).

Capillary RBC flux measurements were conducted using 2PM imaging of RBC-induced shadows within blood plasma labeled by the Alexa-680, allowing the deep imaging into the subcortical white matter down to a depth of ~1.1 mm (*Li et al., 2019*). The mean capillary RBC flux in aged sedentary mice was significantly lower than that in young sedentary mice in the white matter (*Figure 1a*). No change was observed in cortical layers II/III and IV. Importantly, while in young sedentary mice, the mean white matter RBC flux tended to be slightly higher than the gray matter RBC flux, the relationship appeared reversed in aged sedentary mice, where mean RBC flux in the white matter strongly tended to be lower than RBC flux in the gray matter. This suggests that the microcirculation in the white matter is affected more than microcirculation in the gray matter by age-related changes.

We next investigated whether voluntary exercise in aged mice exerts beneficial effects on capillary RBC flux across cortical layers and white matter. We found higher capillary flux in the aged exercise group compared with sedentary controls, which was most prominent in white matter (*Figure 1a*) and to a lesser extent in layer IV, while no change was observed in cortical layers II/III. Cumulative histograms of capillary RBC flux in gray and whiter matter confirmed this finding (*Figure 1b* and *Figure 1—figure supplement 1*). Exercise also tended to increase capillary RBC speed in layer IV of the cortex and the underlying white matter, while no change was detected in layers II/III (*Figure 1—figure supplement 2*). No significant difference in the RBC line-density between aged sedentary and aged exercise groups was found (*Figure 1—figure supplement 3*). The RBC line-density of subcortical white matter was significantly higher than the RBC line-density of gray matter in both aged sedentary and aged exercise groups, consistent with the previous numerical simulation result (*Gould et al., 2017*).

Exercise decreased the coefficient of variation (CV) of capillary RBC flux in subcortical white matter but not the gray matter, suggesting more homogeneous microvascular blood flow in white matter in the aged exercise group (*Figure 1c*). In contrast, we did not find a difference in the CV of capillary RBC flux among the layers in the sedentary group.

Finally, we employed Doppler OCT to test the effect of exercise on the blood flow in larger vessels, particularly the ascending venules in the cortex. We performed a linear regression analysis between the venular blood flow and the logarithmic vessel diameter (*Figure 1d*). As expected, a strong correlation was found between the venular flow and the vessel diameter in both groups, consistent with the previous observation of positive correlation between venular flow speed and the vessel diameter in mice (*Santisakultarm et al., 2012*). Importantly, the regression slope for the aged exercise group was steeper than that for the sedentary group (p = 0.005, analysis of covariance). Blood flow in ascending venules was significantly larger (by ~46%) in the aged exercise group compared to sedentary controls (*Figure 1e*). This result agreed with the improved capillary RBC flux in the white matter with exercise. Aged exercise group showed a larger variation of the venular flow than aged sedentary group, possibly because of the smaller sample size compared with aged sedentary group. The variation may also be related to the variation of the vessel diameter (*Figure 1d*) although no significant differences in the mean and standard deviation of vessel diameter between two groups were found (aged sedentary group: 25.1 ± 3.6 μm vs. aged exercise group: 27.6 ± 3.6 μm, mean ± standard deviation).

The capillary RBC flux in subcortical white matter was moderately negatively correlated with the average daily running distance (i.e., exercise intensity; $R^2$ = 0.31). No strong correlation was found between other measured parameters and the exercise intensity (*Figure 1—figure supplement 4*).

## Aging-induced reduced microvascular oxygenation in deeper cortical regions was rescued by 5 months of voluntary exercise

We next asked whether aging- and exercise-induced changes in capillary RBC flux affected the microvascular oxygenation as well. We performed two-photon phosphorescence lifetime imaging using our two-photon microscope and a phosphorescent oxygen-sensitive probe (Oxyphor 2P) to examine variations of capillary mean $pO_2$ in both aged sedentary and aged exercise groups as a function of cortical depth (*Figure 2a*). We have previously shown that the resting-state capillary mean $pO_2$ gradually increased from layer I to IV by ~6 mmHg in young adult mice (3–5 months old) of the same strain and sex (*Li et al., 2019*). In contrast, the aged sedentary mice here exhibited a different pattern, with the capillary mean $pO_2$ reaching a plateau in layers II/III (capillary mean $pO_2$, layer I: 42 ± 1 mmHg, layers II/III: 46 ± 1 mmHg, layer IV: 45 ± 2 mmHg).

The aged exercise group showed an overall increase in capillary mean $pO_2$ in comparison with the aged sedentary group across all cortical layers (*Figure 2a*). The $pO_2$ increase in cortical layer IV was

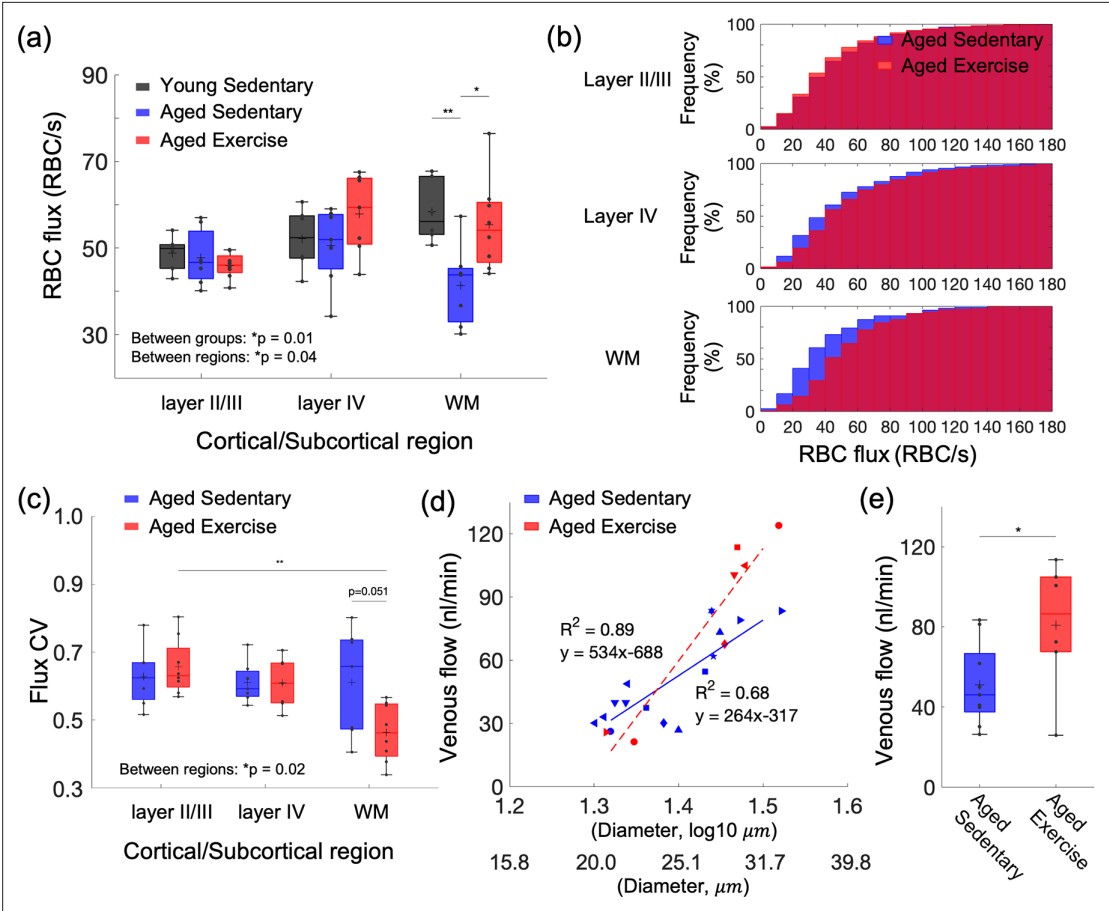

**Figure 1.** Aging- and exercise-induced alterations in cerebral microcirculation. (**a**) Capillary red-blood-cell (RBC) flux across cortical layers II/III and IV, and subcortical white matter in young sedentary, aged sedentary, and aged exercise groups. (**b**) Cumulative histograms of capillary RBC flux in the gray and white matter in aged sedentary and aged exercise groups. (**c**) The coefficient of variance (CV) of capillary RBC flux across cortical layers II/III and IV, and subcortical white matter in each animal group. (**d**) Venular flow versus vessel diameter. Different symbols represent different animals. The red dashed and blue solid line is the best fit result of each linear regression for aged sedentary and aged exercise groups, respectively. (**e**) Mean venular flow in ascending venules in (**d**) in aged sedentary and exercise groups. The measured flow values from all the venules were first averaged to obtain the mean flow for each mouse. The mean flow values for each animal group were then obtained by averaging over mice from that group. The data in (**a**) are from 264, 142, and 168 capillaries in six mice in the young sedentary group, 921, 486, and 112 capillaries in seven mice in the aged sedentary group, and 1046, 465, and 238 capillaries in eight mice in the aged exercise group, in cortical layers II/III, IV, and the white matter, respectively. The data in (**d**) and (**e**) are from 14 and 7 ascending venules in 9 and 6 mice in the aged sedentary and aged exercise groups, respectively. Statistical analysis was carried out using two-way analysis of variance (ANOVA) with post hoc Tukey's in (**a**) and (**c**) and Student's *t*-test in (**e**). *p < 0.05; **p < 0.01. Additional details on boxplots and animals excluded from the analyses are provided in the Supplementary document.

The online version of this article includes the following source data and figure supplement(s) for figure 1:

**Source data 1.** Capillary red-blood-cell (RBC) flux measured in young and aged mice.

**Source data 2.** Capillary red-blood-cell (RBC) flux coefficient of variance (CV) measured in aged mice.

**Source data 3.** Venous flow measured in aged mice.

**Figure supplement 1.** Histograms of capillary red-blood-cell (RBC) flux in the gray and white matter in each animal group.

**Figure supplement 2.** Exercise-induced alterations in capillary red-blood-cell (RBC) speed.

**Figure supplement 3.** Exercise-induced alterations in capillary red-blood-cell (RBC) line-density.

**Figure supplement 4.** Correlations between the capillary red-blood-cell (RBC) flux and coefficient of variance (CV), and the running activity.

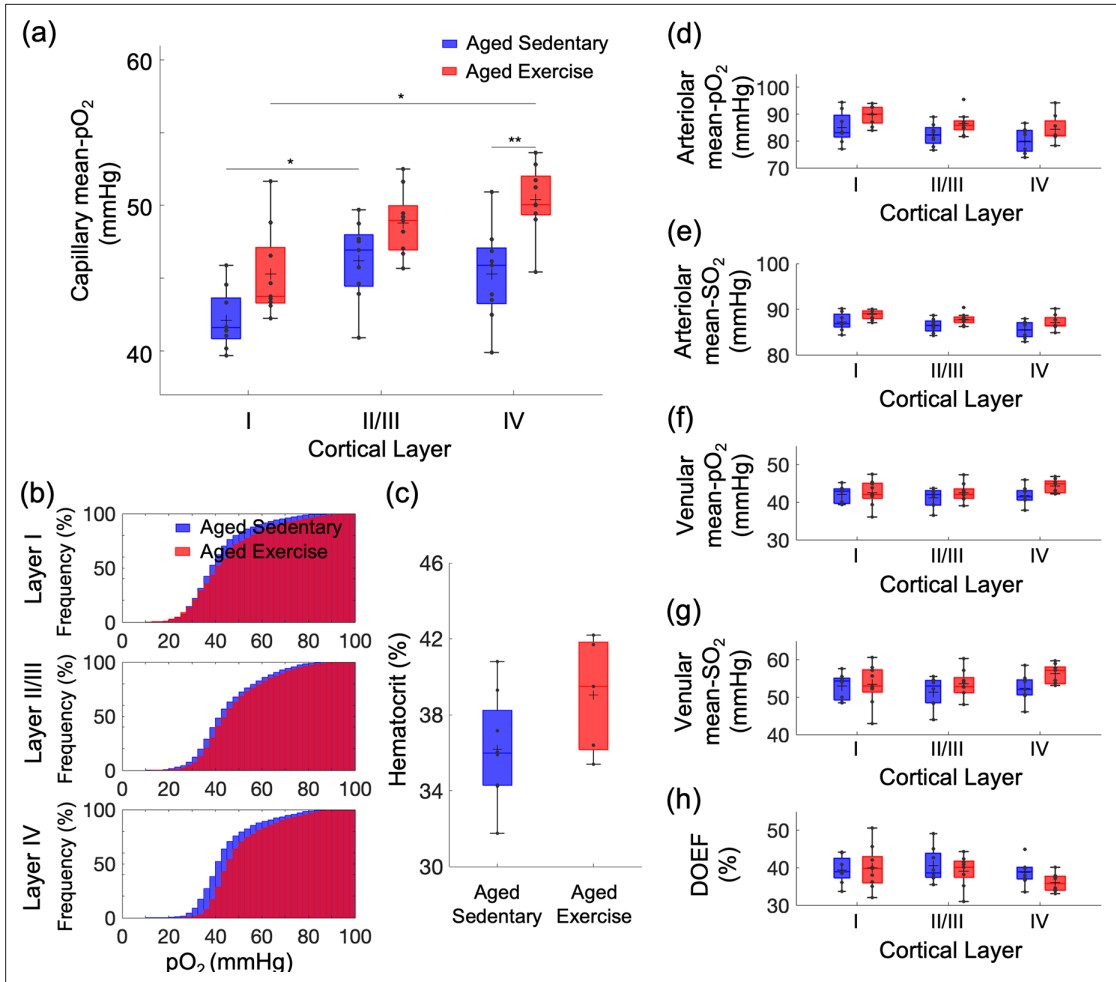

**Figure 2.** Exercise-induced changes in microvascular pO₂ across cortical layers in old mice. (**a**) Capillary mean pO₂ across cortical layers in aged sedentary controls and exercising mice. (**b**) Cumulative histograms of capillary pO₂ in layers I, II/III, and IV. (**c**) The mean Hct levels from aged sedentary ($n = 8$) and aged exercise groups ($n = 5$). (**d, e**) Intravascular pO₂ and SO₂ in the diving arterioles across cortical layers I–IV in aged sedentary (blue boxplots) and aged exercise (red boxplots) groups, respectively. (**f, g**) Intravascular pO₂ and SO₂ in the surfacing venules across cortical layers I–IV in aged sedentary (blue boxplots) and aged exercise (red boxplots) groups, respectively. (**h**) Depth-dependent OEF in aged sedentary (blue boxplots) and aged exercise (red boxplots) groups. The analysis in (**a**) and (**b**) was made with 1224, 2601, and 922 capillaries across $n = 9$ mice in aged sedentary group and 1334, 2840, and 1078 capillaries across $n = 9$ mice in aged exercise group in cortical layers I, II/III, and IV, respectively. The analysis in (**d–h**) was made with 13 arterioles and 12 venules from $n = 9$ mice in aged sedentary group and 14 arterioles and 12 venules from $n = 9$ mice in aged exercise group. Statistical analysis was carried out using two-way analysis of variance (ANOVA) with post hoc Tukey's in (**a**) and (**d–h**) and Student's $t$-test in (**f**). *$p < 0.05$; **$p < 0.01$. Additional details on boxplots and exclusions are provided in the Supplementary document.

The online version of this article includes the following source data and figure supplement(s) for figure 2:

**Source data 1.** Capillary pO₂ measured in aged mice.

**Source data 2.** Blood hematocrit level measured in aged mice.

**Source data 3.** Arterial (and venous) pO₂ measured in aged mice.

**Figure supplement 1.** Histograms of capillary pO₂ in layers I, II/III, and IV.

**Figure supplement 2.** Capillary mean pO₂ versus capillary red-blood-cell (RBC) flux in the mouse cortex.

more pronounced compared to the other layers. The distribution of capillary pO₂ in the aged exercise group also shifted toward higher pO₂ compared with aged sedentary controls (*Figure 2b* and *Figure 2—figure supplement 1*), which was again more pronounced in layer IV. Consistent with our previous report (*Li et al., 2019*), we also found a strong positive correlation between the mean pO₂ and mean capillary RBC flux in both groups (*Figure 2—figure supplement 2*).

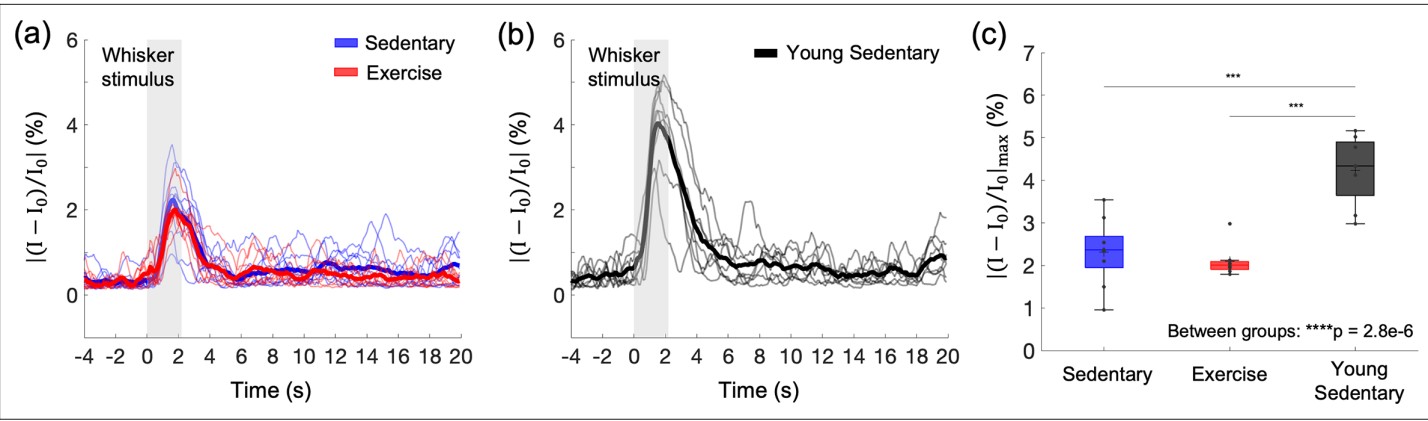

**Figure 3.** Effects of aging and exercise on functional hemodynamic response. Optical intrinsic signal time courses in the whisker barrel cortex of individual old mice in aged sedentary (blue; n = 9) and aged exercise (red; n = 8) mice (**a**), and young (7 months old) sedentary mice (n = 8) (**b**). Thick curves represent averages. (**c**) Average changes in the peak intensity in aged sedentary, aged exercise, and young groups. One-way analysis of variance (ANOVA) with Tukey post hoc test. **p < 0.01; ***p < 0.001; ****p < 0.0001. Please see Supplementary document for exclusions.

The online version of this article includes the following source data and figure supplement(s) for figure 3:

**Source data 1.** Peak hemodynamic response amplitude measured in young and aged mice.

**Figure supplement 1.** Differences in the latency (time to peak) of stimulus-induced hemodynamic response between aged sedentary, aged exercise, and young sedentary mice.

The increase in the capillary $pO_2$ could be in part due to an increase in Hct level due to exercise (*Moeini et al., 2020*). We observed a moderate but statistically insignificant increase in Hct in the aged exercise group compared with aged sedentary mice (*Figure 2c*). No difference in the capillary RBC line-density between two groups was found (*Figure 1—figure supplement 2*). No correlation between the measured parameters (mean $pO_2$ and Hct) and the exercise intensity was found (data not shown).

The $pO_2$ and the oxygen saturation of hemoglobin ($SO_2$) in the diving cortical arterioles and ascending venules across cortical layers were lower in the aged sedentary mice (*Figure 2d–g*) than what we have reported in young adult mice (*Li et al., 2019*), and in agreement with the $pO_2$ values in old mice previously reported by others (*Moeini et al., 2018*). Importantly, both $pO_2$ and $SO_2$ tended to increase in all cortical layers in aged exercise group compared with aged sedentary group, particularly in layer IV, consistent with the largest changes in the capillary RBC flux and mean $pO_2$ observed in the deeper cortical layers. Consequently, the depth-dependent oxygen extraction fraction (DOEF) decreased in the aged exercise group compared with the aged sedentary controls, with potentially the largest decrease in the layer IV (*Figure 2h*).

## Cortical hemodynamic response to functional activation was reduced by aging but not altered by exercise

The subtle impact of exercise on the gray matter microvascular perfusion and oxygenation led us to question whether it has effects on cortical hemodynamic response. We assessed the effects of aging and exercise on the hemodynamic response to the whisker stimulation using optical intrinsic signal imaging (OISI) (*Figure 3*). Optical intrinsic signal time courses of mice in the aged sedentary and aged exercise groups were obtained (*Figure 3a*) and compared with those of younger (7 months old) sedentary mice (*Figure 3b*). The peak amplitudes of the hemodynamic response were significantly smaller in aged mice compared with young sedentary mice regardless of the exercise status (*Figure 3c*). However, the exercise did not affect the peak response amplitude. Response latency (i.e., time to peak after stimulus onset) also did not differ among the groups (*Figure 3—figure supplement 1*). No correlation between the measured parameters (the peak response amplitude and the response latency) and the exercise intensity was found (data now shown).

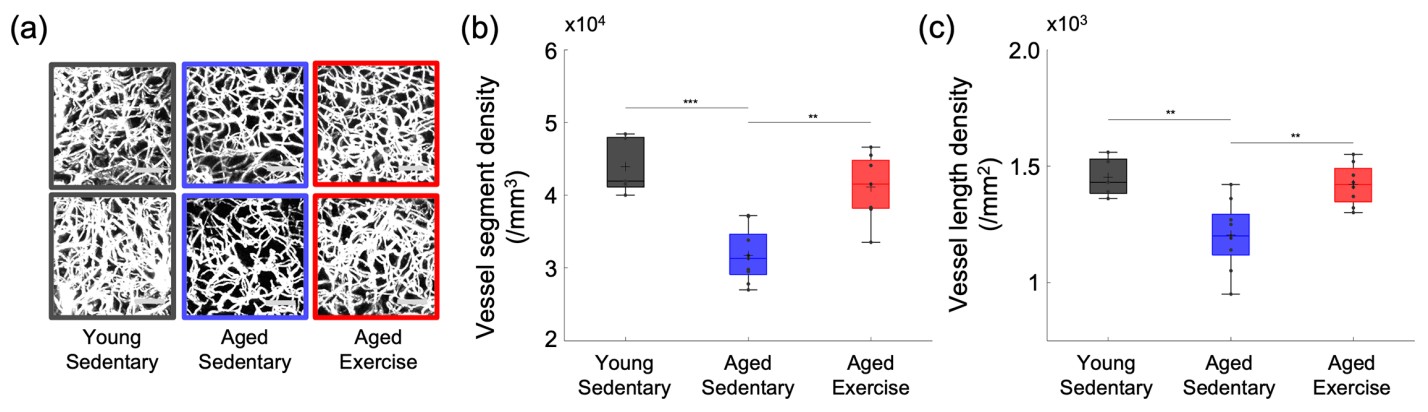

**Figure 4.** Cortical microvascular density in aged mice. (**a**) Representative maximum intensity projection (MIP) images of the three-dimensional angiograms of three mice in the young sedentary group, three mice in the aged sedentary group, and three mice in the aged exercise group, over the cortical depth range from 50 to 400 µm, and 200 × 200 µm² field of view (FOV). Scale bars: 50 µm. (**b**) Vessel segment density and (**c**) vessel length density of cortical capillaries from young sedentary (*n* = 5 mice), aged sedentary (*n* = 9 mice), and aged exercise (*n* = 8 mice) groups. Student's *t*-test. **p < 0.01; ***p < 0.001. Please see Supplementary document for exclusions.

The online version of this article includes the following source data for figure 4:

**Source data 1.** Cortical capillary segment/length density measured in young and aged mice.

## Cortical microvascular density is significantly larger in the aged exercise group

To explore whether aging and excise induces structural changes in the cerebral microvasculature, we segmented three-dimensional stacks of the cortical microvasculature and obtained their mathematical graph representations (*Tsai et al., 2009*). Representative maximum intensity projection images of the microvascular stacks show denser cortical microvascular networks of mice from the young sedentary and aged exercise groups compared to those of the aged sedentary controls (*Figure 4a*). Both young sedentary and aged exercise mice had significantly higher microvascular segment and length density

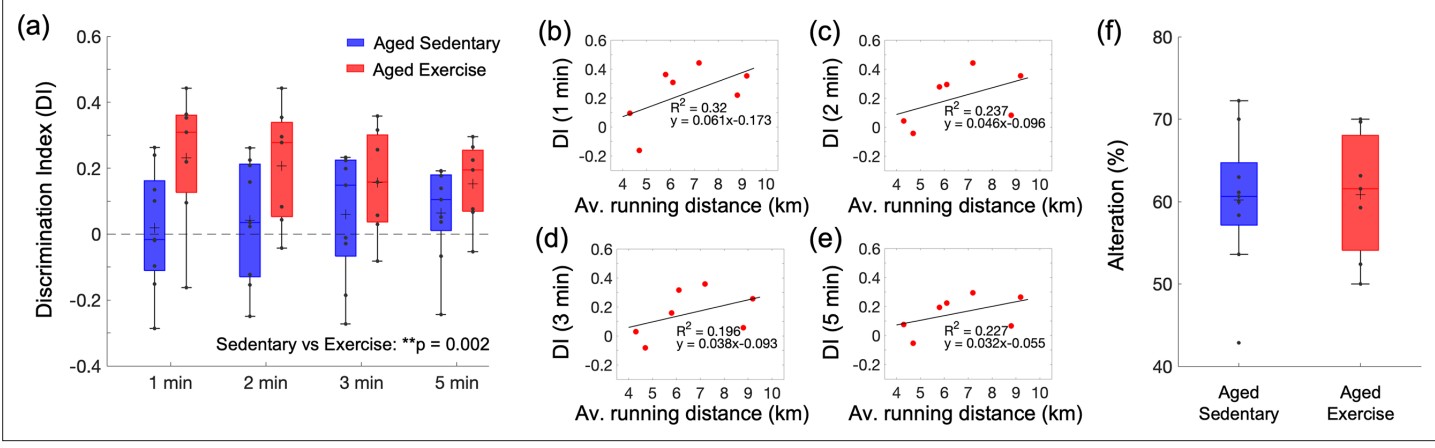

**Figure 5.** Effect of exercise in the old mice on cognitive performance. (**a**) Discrimination index (DI) scores in novel object recognition test (NORT), calculated with four different exploration time periods in the aged sedentary (*n* = 9) and aged exercise (*n* = 7) groups. The calculated DI values at each time interval were subsequently averaged across animals. (**b–e**) Correlations between the daily average running distance and four different DI scores: 1, 2, 3, and 5 min, respectively. (**f**) Spontaneous alteration scores in the Y-maze test in the aged sedentary (*n* = 9) and aged exercise (*n* = 7) groups. Statistical analysis was carried out using two-way analysis of variance (ANOVA) with post hoc Tukey's in (**a**) and Student's *t*-test in (**f**). **p < 0.01. Please see Supplementary document for exclusions.

The online version of this article includes the following source data for figure 5:

**Source data 1.** Behavioral scores from novel object recognition test (NORT) and Y-maze test measured in aged mice.

in the cortical layers I–IV, compared with their aged sedentary mice (*Figure 4b, c*). Segment and length density did not correlate with running activity (data not shown).

## Exercise improves short-term spatial memory performance

In the novel object recognition test (NORT), mice from the aged exercise group spent more time exploring the novel object than the familiar one leading to significantly higher discrimination index (DI) scores across four different time intervals than aged sedentary mice (*Figure 5a*). Interestingly, the DI score for each time interval correlated with the average daily running distance (*Figure 5b–e*). In contrast, aged sedentary and aged exercise groups did not differ in the Y-maze test performance (*Figure 5f*).

## Discussion

In this study, we observed depth-dependent decreases in cerebral microvascular perfusion and oxygenation in aged sedentary mice. The decrease was mitigated by 5 months of exercise in a depth-dependent manner. The key findings include (1) age-associated reduction in capillary RBC flux in the white matter that was moderated by exercise; (2) a decrease in the mean capillary $pO_2$ in cortical layer IV with aging and the overall improvement in capillary $pO_2$ with exercise, with a particularly pronounced $pO_2$ increase in layer IV; (3) an increase in cortical microvascular density with exercise; and (4) improvement of short-term memory function with exercise.

We first assessed the spatial distribution of capillary RBC flux across the cortical and subcortical regions and how it responds to aging and exercise. In aged sedentary mice, the capillary RBC flux in the subcortical white matter tended to be lower than that in the gray matter (*Figure 1a*). Young sedentary mice showed the opposite trend, consistent with our previous finding in anesthetized young adult mice (3–5 months old) showing a higher capillary RBC flux in the subcortical white matter compared to the gray matter (*Li et al., 2020*). Importantly, we observed a large discrepancy in the capillary RBC flux between young and aged sedentary mice in the white matter with preserved capillary RBC flux in the superficial gray matter. Cerebral white matter is known to be more susceptible to hypoperfusion compared to the gray matter, potentially because it is located further downstream with respect to the arterial blood supply (*Gunning-Dixon et al., 2009*; *Markus et al., 2000*). White matter vulnerability to ischemic injury increases with age (*Baltan et al., 2008*), and white matter lesions and lacunar infarcts are common in elderly people with CSVD and hypertension (*Breteler et al., 1994*; *van Swieten et al., 1991*). The vulnerability of white matter to such pathological conditions could be related to our observation of a severe decrease in blood flow in the white matter during normal aging.

Five months of exercise restored the spatial distribution trend of capillary RBC flux close to its distribution in the young sedentary mice (*Figure 1a*). Exercise was associated with a decreased CV of the capillary RBC flux in subcortical white matter, suggesting more homogeneous microvascular blood flow in that region that could potentially facilitate improved oxygen delivery to tissue. Increased capillary perfusion in the white matter was accompanied by increased blood flow in ascending venules, as measured by Doppler OCT (*Figure 1d, e*). In contrast to smaller ascending venules having their branches being distributed mostly over the upper cortical layers, large venules likely extend further to subcortical white matter where the increased capillary perfusion was observed (*Duvernoy et al., 1981*; *Kirst et al., 2020*; *Xiong et al., 2017*). Venules smaller than 20 μm in diameter were excluded from analysis as the small venules tend to have slower flow and may have smaller absolute changes in the blood flow compared to larger venules with higher flow and thus the flow change may not be easily distinguishable due to the relatively low precision of the speed measurement based on Doppler OCT (±0.7 mm/s, see Spectral-domain OCT for more details) (*Fan et al., 2020*; *Santisakultarm et al., 2012*). The increased venous flow in the aged exercise group could be associated with an increased arterial flow, possibly affected by exercise-induced changes in heart strength, brain blood flow regulation, and vascular anatomy. However, flow measurement in arterioles could not be performed with our OCT setup because the flow in many diving arterioles exceeded the maximum measurable limit of the flow measurement by Doppler OCT.

Previous studies in mice reported conflicting results regarding the effects of exercise on cortical microcirculation (*Dorr et al., 2017*; *Falkenhain et al., 2020*; *Lu et al., 2020*). However, these observations have been limited to the blood flow in gray matter, which was shown in this study to be less

responsive to exercise than that in deeper brain regions. Human MRI data acquired after short-term (1 week) exercise showed that exercise induced a selective increase in hippocampal CBF with no or negligible changes in the gray matter CBF (*Steventon et al., 2021*). Our findings suggest that cerebral subcortical microcirculation is more responsive to both age-related and exercise-induced changes than cortical microcirculation.

The change in cortical microvascular oxygenation was also depth dependent. In aged sedentary mice, the mean capillary $pO_2$ increased from layer I to II/III and reached a plateau in layer II/III (*Figure 2a, b*). A different trend was observed in young sedentary mice, which showed a gradual increase in the mean capillary $pO_2$ from layer I to IV (*Li et al., 2019*). In contrast to aged sedentary controls, mice from the aged exercise group showed a similar trend to young sedentary mice, with a pronounced increase in the mean capillary $pO_2$ in layer IV among all assessed cortical layers. In the mouse somatosensory cortex, layer IV exhibits the highest neuronal and capillary densities and the strongest staining for the cytochrome c oxidase, potentially implying the highest oxidative demand during activation and/or at rest throughout all cortical layers (*Blinder et al., 2013*; *Lefort et al., 2009*). In an immunohistochemical study performed using an anti-Glut-1 antibody, high plaque load and decreased blood vessel density were observed, particularly in layer IV of the somatosensory cortex in aged, 18-month-old transgenic AD mice, while younger AD mice did not demonstrate any difference compared to the wild-type mice (*Kuznetsova and Schliebs, 2013*). The decrease in capillary mean $pO_2$ in layer IV in aged sedentary mice could be associated with the larger mismatch in the oxygen delivery and consumption compared with the more superficial cortical layers. However, due to technical limitations, we did not assess the microvascular oxygenation in the deeper cortical layers and subcortical white matter. Because RBC flux exhibits the largest discrepancy between sedentary aged and young mice in the subcortical white matter, it is possible that intravascular $pO_2$ also exhibits the largest discrepancy in this brain region.

Voluntary exercise significantly improved microvascular oxygenation compared to sedentary controls, particularly in layer IV. Similar to our finding of capillary RBC flux improvement due to exercise mostly in the white matter, exercise differentially affected cortical intravascular oxygenation, with the largest increase in the deeper layers and restored the spatial distribution trend of capillary $pO_2$ across cortical layers close to its distribution in the young sedentary mice. However, since the difference in both the Hct level and RBC line-density between the two groups was not significant (*Figure 2c* and *Figure 1—figure supplement 3*), factors other than increased Hct may be involved in the observed depth-dependent $pO_2$ increase in the gray matter. Interestingly, we found that the RBC line-density in both groups of aged mice was significantly higher in the subcortical white matter than in the gray matter. This was not observed in young adult mice (*Li et al., 2020*). However, this finding is consistent with previous simulation results that showed higher Hct levels in deep-reaching penetrating arterioles compared to arterioles whose branches connect to the capillary bed closer to the surface due to the plasma skimming effect (*Gould et al., 2017*).

Consistent with the small effect of exercise on cerebral microcirculation and oxygenation in superficial cortical areas, we observed no change in the relative peak amplitude or the latency of stimulus-induced hemodynamic response with exercise, assessed with OISI at 570 nm, which emphasizes the intrinsic signal originating from cerebral blood volume changes in the superficial cortical layers (*Figure 3* and *Figure 3—figure supplement 1*; *Malonek et al., 1997*; *Tian et al., 2011*). Young sedentary mice (7 months old) showed a significantly larger relative response amplitude than aged mice, consistent with the reduced cerebrovascular reactivity with age in healthy adults and rodents (*Bálint et al., 2019*; *Barnes, 2015*; *Cai et al., 2023*; *Jessen et al., 2017*; *Seker et al., 2021*). The baseline CBF can have a strong effect on the magnitude of the hemodynamic response (*Buxton et al., 2004*; *Corfield et al., 2001*). While no statistically significant difference in the mean capillary RBC flux in the gray matter was found between the aged sedentary and aged exercise groups (*Figure 1a*), the aged exercise group had larger capillary density in the gray matter (*Figure 4b*), suggesting that cortical blood perfusion was possibly also higher in the aged exercise group. This was further supported by significantly higher blood flow in ascending venules in the same group (*Figure 1d and e*). Therefore, our data imply that with a similar level of the relative response amplitude between two groups, the aged exercise group potentially had a larger transient blood supply to the tissue during functional hyperemia.

Consistent with our findings, aging-associated decrease and exercise-induced increase in cerebral microvascular density have been observed across brain regions in both humans and rodents (*Ding et al., 2006*; *Morland et al., 2017*; *Riddle et al., 2003*). However, other studies showed no improvements in cerebral microvascular structure with regular exercise in the mouse sensorimotor cortex (*Cudmore et al., 2017*; *Dorr et al., 2017*; *Falkenhain et al., 2020*). Exercise can affect the brain through different mechanisms across different brain regions at various points in the lifespan (*Stillman et al., 2020*). Although it is unclear whether the difference in age or brain region (or both) contributed to the inconsistent results with some of the previous observations, we found significant morphological changes in the microvascular morphometric parameters due to chronic exercise in the somatosensory cortex of 20-month-old mice (*Figure 4*). Based on the observed changes in the microvascular RBC flux and $pO_2$, we anticipate that an even greater discrepancy in the capillary density between the two groups of mice may be observed in deeper cortical layers and subcortical white matter. It is compelling to hypothesize that higher microvascular density is one of the major contributors to the larger microvascular perfusion and oxygenation observed in mice from the aged exercise group. However, it is not clear whether the larger capillary density in the aged exercise group is mostly due to increased angiogenesis or decreased pruning of the capillaries in comparison with the aged sedentary group. In addition, if increased angiogenesis is present in the aged exercise group, it will be important to better understand the contribution of the new capillaries to oxygen delivery to tissue.

Increased cerebral perfusion and oxygenation with exercise were accompanied by improved spatial short-term memory function, as evaluated by NORT (*Figure 5a*). The perirhinal cortex plays an important role in object recognition memory. It receives sensory inputs from its neighboring sensory cortices, such as the somatosensory cortex, where we found significant improvements in vascular function and structure due to exercise in aged mice (*Antunes and Biala, 2012*; *Cohen and Stackman Jr., 2015*). The NORT results depend on the exploration time in the test phase, which is related to the age-dependent decay of the novel object preference (*Traschütz et al., 2018*). Therefore, choosing an adequate time interval to calculate the DI is important to reliably detect novel object recognition, especially in aged mice. In our analysis, the DI score was calculated at four different time intervals: 1, 2, 3, and 5 min. The results showed an increased DI score in running mice compared to aged sedentary mice across all time intervals, which was also positively correlated with both the average and total running distance (*Figure 5b*). In contrast, we did not observe an increase in spontaneous alterations between the arms of the Y-maze (*Figure 5c*). Consistent with our findings, microcirculatory changes in the white matter have been associated with recognition memory function evaluated by NORT in rodents, while no associations were found with spatial memory function assessed with Y-maze test in the same studies (*Blasi et al., 2014*; *Choi et al., 2015*). However, we do not rule out the possibility that exercise may improve spatial memory function and suggest further investigation to evaluate different aspects of spatial memory function by applying other behavioral tests (e.g., Morris water maze test). As previously shown in several studies in mouse models of AD, each memory task depends on a variety of brain regions that can be differentially affected by exercise, which could also produce inconsistent results between different memory tasks (*Kraeuter et al., 2019*; *Winters et al., 2004*).

A recent study reported a positive correlation between the average daily running distance and cerebral tissue oxygenation in 6-month-old mice after three months of exercise (*Lu et al., 2020*). While mice in our aged exercise group had significantly higher white matter capillary RBC flux than aged sedentary mice, the mean capillary RBC flux in the white matter was negatively correlated with the average daily running distance (*Figure 1—figure supplement 4*). Although white matter capillary density was not assessed in this study, lower capillary RBC flux in the white matter could potentially be accompanied with higher capillary density. In this case, capillary RBC flux in the white matter can decrease but cerebral tissue oxygenation in this region can still increase in response to chronic exercise. This result also may be due to a non-proportional relationship between exercise and brain function, which could decrease after reaching the optimal intensity of exercise (*Khakroo Abkenar et al., 2019*). It is possible that 5 months of unrestricted exercise led some (or all) animals to over-exercise, which may adversely affect subcortical capillary blood flow and prevent achieving the maximum benefits of exercise on the microcirculation. Among different optical measurements performed in this study, only the capillary RBC flux in subcortical white matter tended to correlate with the running activity while no correlations were found between the parameters measured in cortical gray matter (e.g., venous flow, capillary $pO_2$, cortical hemodynamic response, and capillary density) and the average running

distance. This could also be attributed to a higher responsiveness of subcortical white matter to exercise compared with the cortical gray matter. Correlations between the measured parameters and the total running distance were analyzed. Because of approximately the same number of days that all mice were running, total and average running distances share a common scaling factor (e.g., number of days) and show nearly identical correlations with all of the parameters measured (data not shown). The present study used female mice because the comparison with our previous studies in young adult female mice and prior data have shown that in comparison to male mice, female mice exhibit higher voluntary running activity that could potentially produce a greater effect on the brain (*Jones et al., 1990*; *Konhilas et al., 2015*; *Rosenfeld, 2017*). However, the effect may vary in male mice as the animal sex could differentially affect the age-related and exercise-induced changes in cerebral microcirculation due to sex-related hormonal differences (*Bisset et al., 2022*; *McMullan et al., 2016*). Although all the groups were subjected to the same housing conditions, isolated housing could be considered as a stressor that can influence the observed changes in the cerebral microcirculation and behavioral function. Aerobic exercise has been shown to enhance cell proliferation in the dentate gyrus regardless of housing condition in young and aged mice (*Kannangara et al., 2011*). Further investigation into the influence of environmental enrichment factors on the age-related and exercise-induced changes will help to identify the mechanisms that underlie these processes.

In the analysis of the capillary RBC flux and capillary $pO_2$, all capillaries identified within the field of view (FOV) were selected and used for analysis without considering their branching orders. In future studies, analysis based on different types of vessels classified by branching order could potentially provide additional insight into capillary blood flow and oxygenation, as we previously showed different characteristics of capillaries with low branching orders located close to precapillary arterioles compared with higher-order capillaries (*Li et al., 2019*).

In conclusion, leveraging our multimodal optical imaging tools, we quantified the changes in microvascular function, structure, and sensory-evoked functional hyperemia in response to aging

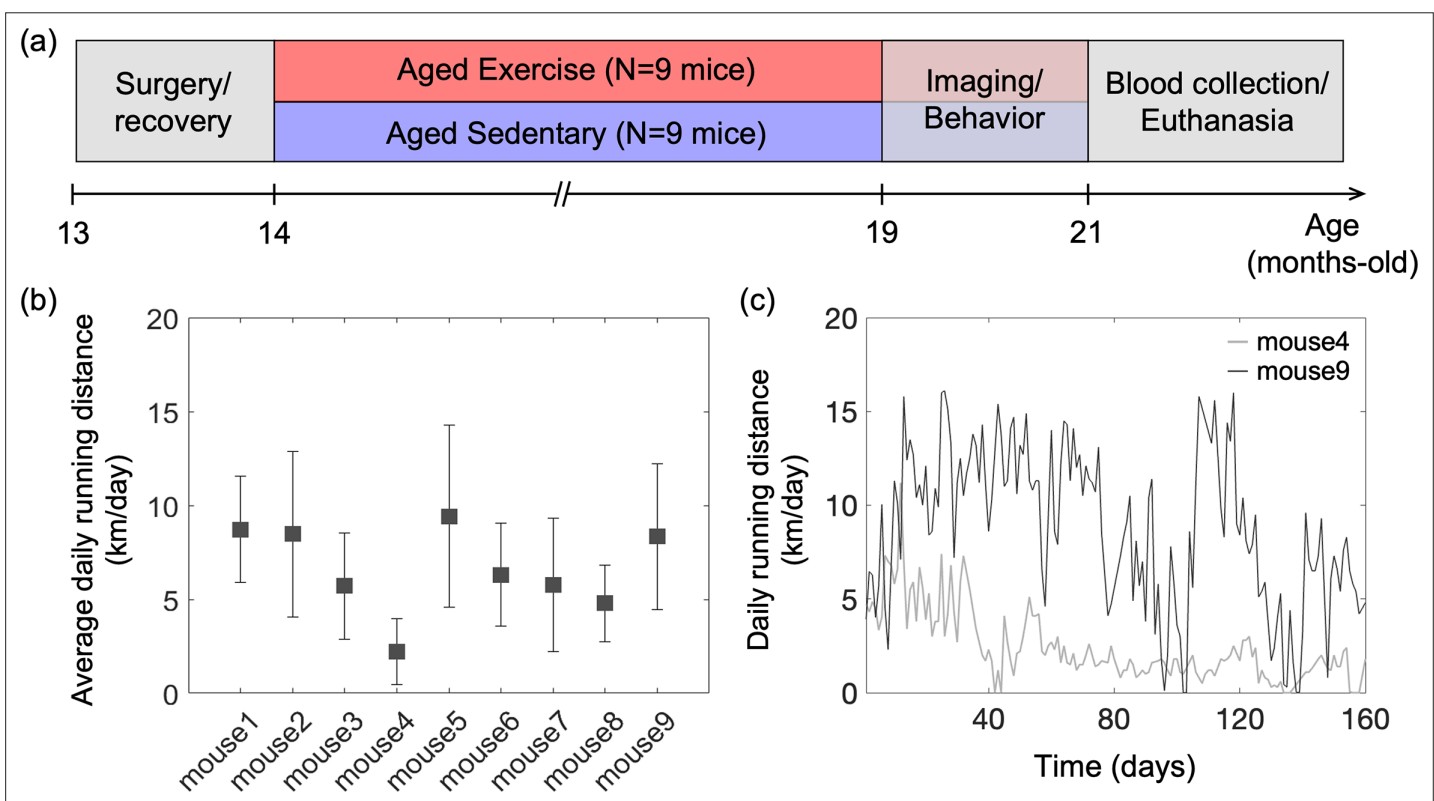

**Figure 6.** Animal preparation and experiment design. (**a**) Timeline of the study. Optical measurements and behavioral testing were performed after 5 months of voluntary exercise when the animals were 19–21 months of age. (**b**) Average daily running distance in km per day for each mouse in the aged exercise group, calculated as the sum of daily running distance divided by a total running period of 5 months. Data are shown as mean ± standard deviation (SD). (**c**) Daily running distance for two representative mice in the aged exercise group across time.

and voluntary exercise in 20-month-old mice. Our results indicate that cerebral microcirculation and oxygenation in deeper cortical layers and subcortical white matter are more susceptible to age-related degeneration, but they are surprisingly more responsive to voluntary aerobic exercise, which induces significant improvements in capillary density, RBC flux, and intracapillary $pO_2$. Improvements in cerebrovascular function and structure are accompanied by rescue of cognitive function. These findings may help us to better understand the patterns and consequences of age-related decline of microcirculation at different cortical depths and subcortical white matter, and how the neurologic effects of aging may be ameliorated by aerobic exercise.

## Materials and methods

### Animal preparation and experimental protocol

All animal surgical and experimental procedures were conducted following the Guide for the Care and Use of Laboratory Animals and approved by the Massachusetts General Hospital Subcommittee on Research Animal Care (Protocol No. 2007N000050). All efforts were made to minimize the number of animals used and their suffering, in accordance with the Animal Research: Reporting in Vivo Experiments (ARRIVE) guidelines. Female C57BL/6N mice (*n* = 18, 12 months old) were obtained from National Institute on Aging colonies. The experimental timeline of the study is shown in *Figure 6a*. A chronic cranial window (round shape, 3 mm in diameter) was implanted on the left hemisphere, centered over the E1 whisker barrel (2.0 mm posterior to bregma and 3.0 mm lateral from the midline) at the age of 13 months. Mice were allowed 4 weeks to recover fully from surgery. Next, mice were randomly divided into two groups (aged exercise: *n* = 9; aged sedentary: *n* = 9), and all mice were housed individually. Cages of mice in the aged exercise group were equipped with wireless running wheels (ENV-047; Med Associates) that could be used by mice at any time. The total number of wheel rotations was recorded daily through a wireless hub device (DIG-807; Med Associates). For each mouse in the aged exercise group, the wheel running activity was characterized based on the intensity of exercise, defined as the average daily running distance over 5 months (*Figure 6b, c*). After 5 months of voluntary exercise (or standard housing for the aged sedentary controls), mice underwent restraint training for awake imaging. Mice in the aged exercise group were continuously housed in cages with running wheels during both training and imaging weeks. All mice were gradually habituated to longer restraint periods, up to 2 hr. They were rewarded with sweetened milk every ~15 min during training and imaging. Optical measurements were performed between 19 and 21 months of age. The measurements were performed during 5–6 weeks in the following order: two-photon phosphorescence $pO_2$ imaging, 2PM angiography, OISI, 2PM imaging of capillary RBC flux, and Doppler OCT. Seven to ten days of a break was given between measurements in each animal. Finally, behavioral tests were performed, and blood samples were collected from the animals for hematocrit (Hct) measurements before the animals were euthanized.

Some measurements in the old mice were compared with similar measurements in the young adult mice. Female C57BL/6N mice, 6 months old (*n* = 8) for OISI and 3 months old for capillary RBC flux imaging (*n* = 6) and microvascular imaging (*n* = 5) were used in this work. Chronic cranial window implantation, animal recovery from the surgery, and restraint training were performed following the same protocols as in the old mice.

### Multimodal optical imaging in awake mice at rest

In this study, we employed our previously described home-built multimodal imaging system that features multiple optical imaging capabilities, including 2PM, OCT, and OISI (*Li et al., 2020*; *Sakadžić et al., 2010*; *Yaseen et al., 2015*). *Figure 7a* shows a schematic of the multimodal imaging system. All optical measurements were made in the head-restrained awake mice at rest. The measurements were conducted while mice were resting on a suspended soft fabric bed in a home-built imaging platform. An accelerometer was attached to the suspended bed to monitor the signals induced by animal motion. Data affected by the motion were rejected during data processing based on the signal generated by the accelerometer. The data acquired when the accelerometer reading exceeded the threshold value, which was determined empirically by comparing signals obtained during stationary and movement periods, were rejected from the analysis. During the experiments, animal behavior was continuously monitored using a web camera (LifeCam Cinema; Microsoft) with a LED illumination at

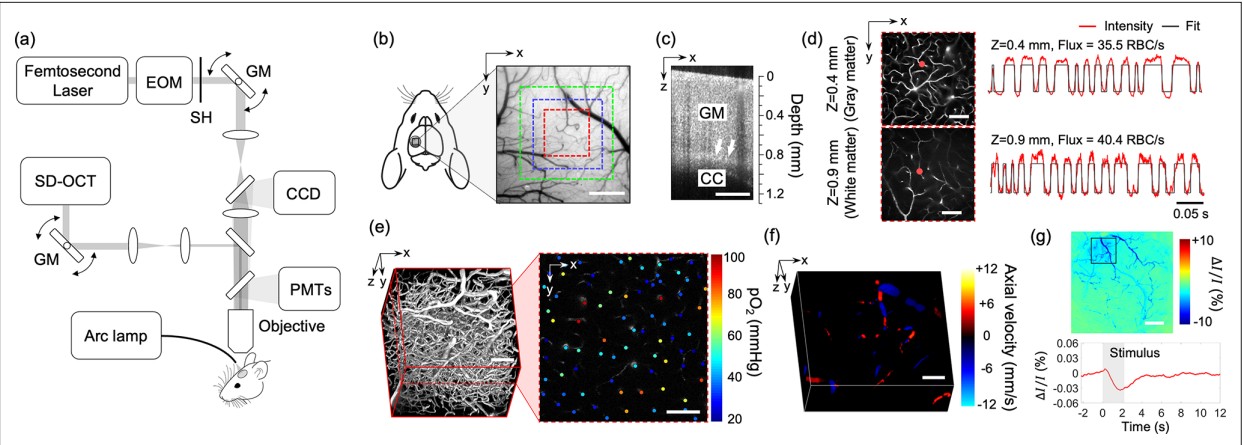

**Figure 7.** Experimental setup and imaging protocols. (**a**) Schematic of our home-built multimodal optical system featuring primary components of the system. A 50-kHz spectral-domain optical coherence tomography (OCT) system was designed to partially share the imaging optics with the two-photon microscope. A Hg:Xe arc lamp in combination with a CCD camera was used for optical intrinsic signal imaging (OISI). EOM: electro-optic modulator, SH: shutter, GM: galvanometer mirror pair. (**b**) A CCD image of brain surface vasculature in the mouse barrel cortex showing the regions of interest (ROIs) where various optical measurements were performed (red ROI: capillary red-blood-cell [RBC] flux, pO2, and microvasculature imaging, blue ROI: Doppler OCT imaging, and green ROI: OCT intensity imaging). (**c**) A representative OCT intensity B-scan image extracted from a volumetric OCT image. White arrows indicate the boundary between the gray matter (GM) and the corpus callosum (CC), which appears as a bright band in the image. (**d**) Survey scan images of cerebral microvasculature of the region outlined with the red square in (**b**) obtained by two-photon microscope at two imaging depths (z = 0.4 and 0.9 mm). Two representative fluorescent intensity time courses acquired within the capillaries at the locations indicated by the red dots in the survey angiograms are presented on the right. (**e**) A 3D angiogram of the mouse cortex acquired by the two-photon microscope at the location outlined by the red square in (**b**). One representative 2D plane from the angiogram acquired at a depth of 200 μm showing pO2 measurements from different capillary segments. pO2 values were color coded (in mmHg) and spatially co-registered with the angiogram. (**f**) A 3D Doppler OCT image showing an axial velocity map of the diving vessels at the location outlined by the blue square in (**b**). (**g**) An OIS image of the cranial window obtained by calculating the relative intensity difference between the post-stimulus response image and pre-stimulus baseline. The region of activation is manually selected from the OIS image as indicated by a black square. The lower panel shows a time course of the relative intensity change due to sensory-evoked hemodynamic response induced by a 2-s-long whisker stimulation, averaged over the selected region of interest. Scale bars: 400 μm for (**b**) and (**c**), 100 μm for (**d–f**), and 500 μm for (**g**).

940 nm, and a reward (sweetened milk) was offered every ~15 min. Experiments were terminated if signs of discomfort or anxiety were observed.

## Two-photon microscope for fluorescence and phosphorescence imaging

A two-photon microscope integrated into our multimodal imaging system was used for capillary RBC flux measurements and 3D microvascular angiography (*Li et al., 2019*; *Sakadžić et al., 2010*). A pulsed laser (InSight DeepSee; Spectra-Physics, tuning range: 680–1300 nm, ~120 fs pulse width, 80 MHz pulse repetition rate) was used as an excitation light source. Laser power was controlled by an electro-optic modulator (EOM) (350-160; ConOptics, Inc). The laser beam was focused by a water immersion objective lens (XLUMPLFLN20XW; Olympus) and scanned in the transverse (*X–Y*) plane by a pair of galvanometer mirrors (Saturn 5B; Pangolin Laser System, Inc). The objective lens was translated along the *Z*-axis by a motorized stage (M-112.1EG; Physik Instrumente) to probe different depths ranging from the brain surface to depths beyond 1 mm below the surface. Emitted fluorescence was directed to detectors by an epi-dichroic mirror (FF875-Di01-38.151; Semrock Inc) positioned above the objective, followed by an infrared blocker (FF01-890/SP-50; Semrock Inc). The four-channel detector consists of four photomultiplier tubes (PMTs) paired with emission filters that cover a wide range of emission wavelengths. One detector channel with a PMT exhibiting a red-shifted spectral response (H10770PA-50; Hamamatsu) and a 709/167-nm emission filter was connected to a discriminator (C9744; Hamamatsu) and used for capillary RBC flux measurements. Another channel with a multialkali photocathode PMT (R3896; Hamamatsu) and a 525/50-nm emission filter was used for two-photon angiography.

We employed another home-built two-photon microscope for intravascular pO2 measurements. The second 2PM employs a Ti:Sapphire mode-locked laser (Mai Tai HP; Spectral Physics, tuning range: 690–1040 nm, ~100 fs pulse width, 80 MHz pulse repetition rate) as a light source. The output laser

power delivered to the sample is modulated with EOM (350-105-02; ConOptics Inc). The transverse scanning was achieved by a two-axis 7 mm galvanometer scanner (6210 H; Cambridge Technology). The laser beam was relayed to the back focal plane of the objective through the combination of a scan lens (f = 51 mm, 2x AC508-100-B, Thorlabs) and a tube lens (f = 180 mm, Olympus). The Z-axis movement of the objective was controlled by two translation stages (ZFM2020 and PFM450E; Thorlabs). The phosphorescence emission signal from the sample was filtered by a dichroic mirror (FF875-Di01-38.1X51; Semrock Inc) and an emission filter (FF01-795/150-25; Semrock Inc) before detected by a PMT (H10770PA-50; Hamamatsu) and a photon counting unit (C9744; Hamamatsu).

The objective lens was heated by an electric heater (TC-HLS-05; Bioscience Tools) throughout the measurement to maintain the temperature of the water between the objective lens and the cranial window at 36–37°C.

## Spectral-domain OCT

A low-coherence superluminescent diode with a center wavelength of 1300 nm was used as the light source (S5FC-1018S; Thorlabs). The line scan camera was operated with a 50-kHz acquisition rate (GL2048L; Sensors Unlimited). The sample arm of the system partially shares the imaging optics with the two-photon microscope, but it utilizes separate scanning optics, as shown in *Figure 7a*. The system has an axial resolution of 10 μm in tissue. The transverse resolution is 7 μm using a ×10 objective lens (Mitutoyo Plan Apo NIR; Edmund Optics). The incident optical power on the sample was 5 mW. For OCT Doppler measurements, the maximum measurable flow speed without phase wrapping was ±12 mm/s when the direction of flow was parallel to the OCT beam and the minimum detectable velocity was ±0.7 mm/s which was determined by the phase noise of the system, measured as ±0.2 radians.

## OISI system

A Hg:Xe arc lamp (66883; Newport) was used in combination with a band-pass filter (570/10 nm) for OISI. A ×4 objective lens (XLFLUOR4X/340; Olympus) was used to achieve a wide FOV that covered the entire cranial window. Two-dimensional images of the cranial window were acquired using a CCD camera with 100ms exposure time (acA1300; Basler).

## Measurements of capillary RBC flux, speed, and line-density in gray and white matter

Before imaging, the dextran-conjugated Alexa680 solution (70 kDa, 0.1–0.15 ml at 5% wt/vol in PBS; Thermo Fisher Scientific) was retro-orbitally injected into the bloodstream. *Figure 7b* shows a CCD image of the mouse cranial window positioned on the left somatosensory cortex. A volumetric OCT scan was performed with an FOV of 1 × 1 mm$^2$ over the region shown with a green square in *Figure 7b*. The acquired OCT volume was used to identify the boundary between the gray and white matter and measure the cortical thickness as we previously described (*Figure 7c*; *Li et al., 2020*). After confirming localization of the white matter, 2PM was performed over the same region as for the OCT imaging, but with a smaller FOV of 500 × 500 μm$^2$ indicated by a red square region (*Figure 7b*). The RBC flux measurements in the gray matter were performed at cortical depths of 150 and 400 μm, which correspond to layers II/III and IV. In the white matter, measurements were performed at a depth of 0.9–1.1 mm. Awake imaging lasted less than 2 hr and the flux measurements were performed only at three depths due to the time spent performing the OCT imaging prior to the flux measurements. At each depth, a survey image of the vasculature was acquired by raster scanning the beam across the FOV (*Figure 7d*). Then we manually selected measurement locations inside all the capillary segments identified within the survey image. Capillaries were identified based on their network structure and, in this work, all branching vessels from diving arterioles and surfacing venules were defined as capillaries. The laser beam was parked at each location for 0.9 s and the fluorescence signal was detected in the photon-counting mode. The photon counts were binned into 300-μs-wide bins, resulting in a fluorescence intensity time course with 3000 time points. *Figure 7d* shows representative fluorescence signal transients taken from two measurement locations at different depths. We assessed the RBC flux, RBC speed, and RBC line-density within individual capillaries, extracted from the point-based acquisition of fluorescence signal. The capillary RBC flux was defined as the number of RBCs that passed through the vessel per unit time (RBC/s). Following the procedures described in our previous study (*Li et al.,*

*2020*), the fluorescence signal time course was segmented with a binary thresholding approach (red curve in *Figure 7d*) and RBC flux was calculated by counting the number of valleys in the segmented curve normalized by the acquisition time. The capillary RBC speed was defined as the average speed of the RBCs in the vessel (mm/s). Following our previously described procedures (*Li et al., 2019*), RBC speed for each RBC-passage event (valley) in the segmented curve was estimated as $v = D/\Delta t$, where $D$ is RBC diameter, assumed to be 6 µm, and $\Delta t$ is the width of the valley. For each capillary, the speed values estimated from each of the valleys in the segmented curve were averaged to obtain the mean RBC speed. The capillary RBC line-density measures the fraction of the capillary length occupied by RBCs, which is closely related to the capillary hematocrit. The capillary RBC line-density was calculated as the ratio between the combined time duration of all signal valleys to the total duration of the entire time course.

The variability in the RBC flux among capillaries, which is termed capillary flow heterogeneity, was quantified by calculating the CV of capillary RBC flux in cortical layers II/III–IV and subcortical white matter. Here, CV is defined as the ratio of the STD to the mean of capillary RBC flux.

## Intravascular pO$_2$ imaging and calculation of SO$_2$ and depth-dependent OEF

A phosphorescent oxygen-sensitive probe Oxyphor2P was diluted in saline and retro-orbitally injected before imaging (0.05 ml at ~80 µM) (*Esipova et al., 2019*). The pO$_2$ imaging was performed using 950 nm excitation wavelength and with an FOV of $500 \times 500 \mu m^2$ at the same cortical region where the RBC flux measurements were performed (*Figure 7b*). The measurements were performed at cortical depths from the surface to 450 µm depth with 50 µm interval between depth locations. At each depth, two-dimensional raster scan of phosphorescence intensity was performed to acquire a survey image. Then, we manually selected the measurement locations inside all diving arterioles, surfacing venules, and capillary segments visually identified within the survey image. Next, the focus of excitation laser beam was parked at each selected segment to excite the Oxyphor 2P with a 10-µs-long excitation gate. The resulting emitted phosphorescence was acquired during 290-µs-long collection time. Such 300-$\mu$s-long cycle was repeated 2000 times to generate an average phosphorescence decay curve. The averaged curve was fitted to a single-exponential decay, and the lifetime was converted into absolute pO$_2$ using a Stern–Volmer-like expression obtained from independent calibrations (*Esipova et al., 2019*; *Li et al., 2019*; *Li et al., 2020*; *Şencan et al., 2022*). The SO$_2$ and the DOEF were computed following our previously described procedures (*Li et al., 2019*). Briefly, the SO$_2$ was calculated using the Hill equation based on the measured pO$_2$ and the DOEF was calculated as $\left(SO_{2,A} - SO_{2,V}\right)/SO_{2,A}$, where $SO_{2,A}$ and $SO_{2,V}$ represent the mean SO$_2$ in the diving arterioles and surfacing venules in a given cortical layer, respectively.

## Characterization of morphological changes in cortical capillaries

FITC-dextran (Thermo Fisher Scientific, 70 kDa) was diluted in saline and injected retro-orbitally before imaging (0.05 ml at 5% wt/vol). A three-dimensional imaging of the cortical vasculature was performed with $500 \times 500$ µm$^2$ FOV to cover the same region of interest (ROI) used for capillary flux and pO$_2$ measurements (*Figure 7e*). The microvascular stack was generated by repeatedly acquiring images with the axial steps of 1 µm up to a depth of 400 µm. In each animal, a smaller ROI ($200 \times 200 \mu m^2$) was manually selected to cover the region containing mostly capillaries (e.g., capillary bed area). A 3D microvasculature corresponding to the selected ROI was segmented using a vessel segmentation algorithm, VIDA (*Tsai et al., 2009*), and the number of capillary segments per volume (e.g., vessel segment density) and the average capillary segment length per volume (e.g., vessel segment length density) were calculated. The vessel segment was defined as the part of the vessel between two consecutive branching points.

## Quantitative flow measurements using Doppler OCT

A total of 20 Doppler OCT volumes were continuously acquired with a 750 × 750 µm$^2$ FOV at the region indicated by a blue square in *Figure 7b*. Ten volumes exhibiting minimal motion artifacts were selected and averaged to generate a single Doppler OCT volume. Each Doppler OCT volume was comprised of 300 B-scans, where 3000 A-scans were acquired per B-scan. The Doppler volume yields a three-dimensional map of the *z*-projection of RBC velocity (*Figure 7f*). For each depth slice

in the volume, we measured flow in each surfacing venule over the FOV by estimating the integral of the velocity projection over the vessel cross-section following the protocol previously described (*Srinivasan et al., 2011*). We used 3D angiograms acquired by 2PM (see Intravascular $pO_2$ imaging and calculation of $SO_2$ and depth-dependent OEF for more details) to estimate the diameter of the vessels, used for cross-correlation analysis between the flow and vessel diameter. The measured flow values in each vascular segment were averaged within the depth range of 50–100 µm. Flow data measured in diving arterioles were excluded from analysis due to their much higher flow compared to venular flow that often causes excessive phase wrapping and signal fading (*Koch et al., 2009*).

## OISI imaging of hemodynamic response to functional activation

Two-dimensional CCD images were continuously acquired with a CCD camera for 18 s. Five seconds after the onset of image acquisition, a whisker stimulus was applied at 3 Hz for 2 s. A Picospritzer microinjection device (051-0500-900, Parker Hannifin Inc) was used to deliver air puffs (20 psi) for the stimulation of a whisker pad. A plastic air tube was aimed at the E1 whisker to deliver the air. A total of 20 stimulation trials were repeated with an inter-stimulus interval of 25 s. Among 20 trials, the data acquired during excessive animal motion (~10% of the measurements) were excluded from analysis based on a threshold criterion applied to the accelerometer recordings. Bulk motion artifacts in the acquired CCD images caused by small transverse movement (<~50 µm) were compensated by 2D cross-correlation based motion correction algorithm (*Guizar-Sicairos et al., 2008*). After motion compensation, images were averaged over trials before computing the fractional intensity difference between the response and baseline images (*Figure 2g*) as described in our previous work (*Şencan et al., 2022*).

## Behavioral tests

An NORT was performed to evaluate whether exercise improves short-term memory in old mice. Briefly, mice were allowed to explore two identical objects in a testing arena for 5 min and returned to their home cage for 30 min. Mice were then moved back to the arena with one of the objects replaced with a novel object and allowed to explore the objects for 5 min. Behavioral performance was evaluated by estimating the DI. The DI was defined as exploration time of the novel object normalized by combined exploration time of both objects and was calculated during four different time intervals in each animal: during first 1, 2, 3, and 5 min of the object exploration.

Y-maze test was used to measure the willingness of animals to explore new environments and spatial working memory. The testing occurred in a Y-shaped maze with three arms at a 120° angle from each other. Mice were placed at the center of the maze and allowed to freely explore the three arms for 10 min. The number of arm entries and the number of triads (consecutive entries into three different arms) were measured to calculate the percentage of spontaneous alteration, defined as the ratio of the number of alternating triads to total number of arm entries −2.

## Hct measurements

Hct measurements were carried out by a blood gas analyzer (OPTI CCA-TS2; OPTI Medical Systems) using a B-type cassette. Mice were anesthetized with 2% isoflurane, and then approximately 250 µl of venous blood was collected from the inferior vena cava. The collected blood was aliquoted into two 100 µl capillary tubes. The Hct measurement was performed two times with each blood sample and the average of the two measurements was reported.

## Statistical analysis

Statistical analysis was carried out using *t*-test or analysis of variance (MATLAB, MathWorks Inc). p value less than 0.05 was considered statistically significant. Details about the statistical analysis are provided in the figure legends and text, where relevant. Boxplots show the median value with a black line and the mean value with a plus symbol (+). Each box spans between the 25th and the 75th percentiles of the data, defined as interquartile range (IQR). Whiskers of the boxplots extend from the lowest datum within 1.5 times the IQR of the lower quartile of the data to the highest datum within 1.5 times the IQR of the highest quartile of the data. For each parameter measured in this study, the acquired values were averaged to obtain the mean value for each mouse, and then the mean values were averaged over mice. Sample size (i.e., *n* = 9 mice for aged sedentary and aged exercise groups)

were selected based on the assumption that the most demanding one is to detect 30% difference between the mean capillary $pO_2$ values (coefficient of variance = 0.3, power = 0.8, significance = 0.05).

## Acknowledgements

Support of the grants RF1NS121095, R01NS115401, U24EB028941, U19NS123717, U01HL133362, R00MH120053, and R01AG055413 from the National Institutes of Health, USA, and support from the Rappaport Foundation and Leducq Foundation are gratefully acknowledged. The funders had no role in study design, data collection, and interpretation, or the decision to submit the work for publication.

## Additional information

### Funding

| Funder | Grant reference number | Author |
| --- | --- | --- |
| National Institutes of Health | RF1NS121095 | Sava Sakadžić |
| National Institutes of Health | R01NS115401 | Sava Sakadžić |
| National Institutes of Health | U24EB028941 | Sava Sakadžić |
| National Institutes of Health | U19NS123717 | Anna Devor |
| National Institutes of Health | U01HL133362 | Sava Sakadžić |
| National Institutes of Health | R00MH120053 | Ikbal Şencan-Eğilmez |
| National Institutes of Health | R01AG055413 | Chongzhao Ran |
| Rappaport Foundation | | Eng Lo |

The funders had no role in study design, data collection, and interpretation, or the decision to submit the work for publication.

### Author contributions

Paul Shin, Conceptualization, Data curation, Formal analysis, Supervision, Validation, Investigation, Visualization, Methodology, Writing – original draft, Project administration, Writing – review and editing; Qi Pian, Mohammed Alfadhel, Software, Investigation, Methodology, Writing – review and editing; Hidehiro Ishikawa, Gen Hamanaka, Emiri T Mandeville, Shuzhen Guo, Investigation, Writing – review and editing; Buyin Fu, Investigation, Methodology, Writing – review and editing; Srinivasa Rao Allu, Resources, Methodology, Writing – review and editing; Ikbal Şencan-Eğilmez, Baoqiang Li, Software, Methodology, Writing – review and editing; Chongzhao Ran, Resources, Supervision, Funding acquisition, Writing – review and editing; Sergei A Vinogradov, Resources, Software, Supervision, Funding acquisition, Methodology, Writing – review and editing; Cenk Ayata, Conceptualization, Resources, Supervision, Funding acquisition, Writing – original draft, Writing – review and editing; Eng Lo, Conceptualization, Resources, Supervision, Funding acquisition, Writing – review and editing; Ken Arai, Conceptualization, Supervision, Funding acquisition, Investigation, Methodology, Writing – review and editing; Anna Devor, Conceptualization, Resources, Software, Supervision, Funding acquisition, Methodology, Writing – original draft, Writing – review and editing; Sava Sakadžić, Conceptualization, Resources, Software, Supervision, Funding acquisition, Methodology, Writing – original draft, Project administration, Writing – review and editing

### Author ORCIDs

Paul Shin http://orcid.org/0000-0001-8301-1079
Ikbal Şencan-Eğilmez http://orcid.org/0000-0002-2903-041X

Sergei A Vinogradov http://orcid.org/0000-0002-4649-5534
Anna Devor http://orcid.org/0000-0002-5143-3960
Sava Sakadžić http://orcid.org/0000-0001-6318-1193

### Ethics

All animal surgical and experimental procedures were conducted following the Guide for the Care and Use of Laboratory Animals and approved by the Massachusetts General Hospital Subcommittee on Research Animal Care (Protocol No. 2007N000050). All efforts were made to minimize the number of animals used and their suffering, in accordance with the Animal Research: Reporting in Vivo Experiments (ARRIVE) guidelines.

### Decision letter and Author response

Decision letter https://doi.org/10.7554/eLife.86329.sa1
Author response https://doi.org/10.7554/eLife.86329.sa2

## Additional files

### Supplementary files

- MDAR checklist

### Data availability

Figure 3 - Source Data 1, Figure 3 - Source Data 2, Figure 3 - Source Data 3, Figure 4 - Source Data 1, Figure 4 - Source Data 2, Figure 4 - Source Data 3, Figure 5 - Source Data 1, Figure 6 - Source Data 1, and Figure 7 - Source Data 1 contain the numerical data used to generate the figures.

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
