## [Editor Report]

Using convincing approaches with mice, the authors show that aging is associated with a reduction in microvascular perfusion and oxygenation and that voluntary aerobic exercising restored these parameters, especially in the white matter. This work is of broad interest to medical biologists in the field of cerebrovascular diseases.

---

## [Decision Letter]

**Decision letter after peer review:**

Thank you for submitting your article "Aerobic exercise reverses aging-induced depth-dependent decline in cerebral microcirculation" for consideration by *eLife*. Your article has been reviewed by 3 peer reviewers, including Daniel Henrion as the Reviewing Editor and Reviewer #1, and the evaluation has been overseen by Christian Büchel as the Senior Editor. The following individuals involved in the review of your submission have agreed to reveal their identity: Osama F Harraz (Reviewer #2); Robert H. Cudmore (Reviewer #3).

Essential revisions:

1) The use of female mice only should be justified in the manuscript. The authors should discuss the hormonal effect on blood flow.

2) Figure 3: see comments 3 to 5 by reviewer 2 and comments 1 to 10 by reviewer 3. All the points need to be clearly addressed and the manuscript modified accordingly.

3) Figure 5: the authors explain how whisker stimulation led to a decrease in the baseline of the signal (rev.2, comment 6).

4) Figure 6: The authors should show the same vascular density analysis in the young cohort. See detailed comment # 11 by reviewer 3.

5) The authors need to address the fact that these density measurements are done in superficial cortical areas where they do not see pronounced age-related changes in RBC flux. What might the density of deeper regions be changing with aging and what effect would exercise have (comment 12 / rev. 3)?

6) The authors should include a reference to Cudmore et al. (2017) where they perform awake longitudinal in vivo 2P imaging to determine if aerobic exercise can induce changes in capillary structure (length, tortuosity, diameter), angiogenesis, or rarefication. See details in comment 13 by reviewer 3.

7) The authors need to comment on the fact that they often did not see correlations between daily average running and blood flow changes, IOSI (Figure 5), and density (Figure 6) but do see differences in the novel object but not the Y-maze behaviors.

8) Scatter plots showing individual data points are encouraged

*Reviewer #1 (Recommendations for the authors):*

1. The authors used female mice only without explaining why. Indeed, using female mice is very important as too many studies use only male mice and thus neglect to study females. Nevertheless, ideally, using both male and female mice would be better. As female mice do not have menopause but have a progressive decline in circulating estrogens the authors could discuss the possible link between estrogen and cerebral blood flow in their study. Another way to investigate the role of estrogens would be to use intact and ovariectomized female mice.

2. As mice were housed individually one may suspect a change in cerebral and cognitive function due to the stress of isolation as mice are very social animals. Although all the groups were submitted to the same experimental conditions, the authors could add a statement about this possible limitation in the discussion.

3. Whereas the effect of aerobic exercise is clear in old mice it would be interesting to know if the effect of exercise is equivalent in young mice. In other words, is the flow and pO2 gain equivalent in young and old mice with exercise? Same question with the other parameters.

*Reviewer #2 (Recommendations for the authors):*

This manuscript is well written and the evidence presented is convincing that aerobic exercise helps normalize cerebrovascular function. The effects were more evident in deeper cortical tissues and in the white matter. I have a number of comments/questions for the authors to consider:

1. Why only female mice? age-dependent changes in female mice could be crucial for the observed effects which might vary in male mice. It would be useful to know whether male mice display a similar pattern.

2. In the exercise group, mice had 5 months of access to exercise wheels. Were the running wheels removed from the cages between months 19-21?

3. The data presentation in Figure 3a and 3b doesn't clearly compare RBC flux in young versus aged mice. This is a key observation in the study and should be highlighted.

4. Figure 3e: It is not clear how the authors obtained the best-fit result for the exercise group (dotted line), since scattered points seem to follow.

5. Figure 3f: The legend mentions that the data are from 14 and 7 venules from 9 and 6 mice, respectively. What is exactly shown in Figure 3f? Was the standard deviation calculation based on the number of vessels or animals? There are twice the number of recordings from the sedentary group compared to the exercise, which possibly explains in part the very large standard deviation in 3f.

6. I suggest that the authors explain how whisker stimulation led to a decrease in the baseline of the signal. It might not be clear and I cannot find a clear explanation of the calculation of light intensity (Figure 5).

7. Since only NORT behavioral test showed significant differences between exercise and sedentary groups, I believe that the evidence is too preliminary at this stage and would need more corroborations.

8. Scatter plots showing individual data points are encouraged.

*Reviewer #3 (Recommendations for the authors):*

Their conclusions are in line with the experimental data provided. I am asking for the following revision below.

Figure 3 a/b. Could panels a and b be combined? The way it is requires comparing the same y-axis across two plots. Shouldn't they be one plot? If they cannot be one plot, why not?

Figure 3 a/b shows "RBC Flux (RBC/sec)" but the supplemental figure shows "RBC Speed (mm/sec)". Are these the same measurement? If so, can the units be the same? If they are not the same measurement, the authors should explain the differences.

Is this difference because RBC Flux (RBC/sec) is measured with a parked layer and peak/throughs are counted versus RBC Speed (mm/sec) is measured by a more common line scan along the capillary? If this is the case, it needs to be explained.

Figure 3 – Supplemental 2. It is not clear to me what the "RBC line-density" tells us. I understand the definition, "RBC line-density was calculated as the ratio between combined time duration of all valleys to the total duration of the entire time course". The reader needs to be told how to interpret larger or smaller RBC line-density. Please describe this in more detail and link it to a biological intuition or hypothesis.

Figure 3 d. "Exercise decreased the coefficient of variation (CV) of capillary RBC flux …". Please specify how the std/mean (e.g., CV) was calculated. Was this within a single capillary flow measurement using your parked beam to count the passage of RBCs? Or is it the std/mean across measurements within a region like "layer IV" and a grouping like young or old?

Furthermore, I would like the authors to lay out a hypothesis or intuition regarding potential differences in the CV of RBC flux. What does this CV metric tell us? What does it tell us if it is larger and what does it tell us if it is smaller? They show that exercise decreases the CV of RBC flux in white matter but not in the superficial cortex. How do I interpret the meaning of this, what should I take away from this?

Figure 3 – Supplemental 3. The authors look for correlations between average daily running and the RBC flux. I would like to see this analysis using the total distance run per individual. This could be mentioned in the Results section. This kind of total running distance per individual (versus daily averages) could also be used for analysis in other sections such as "3.4 Cortical microvascular density is significantly larger in the exercise group" and behavior experiments in Figure 7b-e.

Figure 3 f. Venous flow is measured with OCT. Why did the authors not measure arteriole flow in the same manner? For example, arterioles were measured for PO2 in figure 4. One hypothesis is that if exercise increased arteriole flow in the aged group, that would partially explain an increase in venous flow. One confounding parameter here is heart health. If the aged exercise group, simply had stronger flow coming from the heart.

Figure 3 c, 4 b. I would like the authors to consider showing these histograms as cumulative histograms. It may be easier for the reader to visually see the shift in the mean of the distribution (compared to a histogram).

"3.3 Cortical hemodynamic response to functional activation was reduced by aging but not altered by exercise" and "Figure 5". The functional hemodynamics measured with OISI is using a video camera focused on the cortical surface and thus represents sensory stimulus-invoked changes in blood flow in superficial cortical areas. Throughout the manuscript, the findings reveal there are few exercise-induced changes in these superficial cortical areas. The authors need to address this in the text. The fact there is no effect of exercise on the superficial evoked blood flow is not at all surprising given many of the other results.

Figure 6. Can the authors show the same vascular density analysis in the young cohort? The hypothesis is that aging reduces vessel density. This would be verification in that it would replicate what has previously been reported and could act as a baseline to compare and discuss the changes in density seen in aged sedentary versus exercising groups. How far does the increase in density in aged exercising mice bring the capillary density toward the young sedentary condition?

The authors need to address the fact that these density measurements are done in superficial cortical areas where they do not see pronounced age-related changes in RBC flux. What might the density of deeper regions be changing with aging and what effect would exercise have?

I want the authors to consider including a reference to Cudmore et al. (2017) where they perform awake longitudinal in vivo 2P imaging to determine if aerobic exercise can induce changes in capillary structure (length, tortuosity, diameter), angiogenesis, or rarefication. This study by Cudmore et al. was performed in young adult mice and they conclude aerobic exercise (voluntary wheel running) does not alter capillary density. It would be interesting to comment on their results (no effect) and your results in aged and exercising mice (big effect).

Cudmore RH, et al. (2017) Cerebral vascular structure in the motor cortex of adult mice is stable and is not altered by voluntary exercise. J Cereb Blood Flow Metab, 37(12):3725-3743. doi: 10.1177/0271678X16682508.

Figure 7. In the behavioral experiments it was found that exercise changed novel exploration but not the Y-maze test. The authors need to comment on why they think this is the case. I guess exercise changes short-term memory (novel object) versus Y-maze (what does the Y-maze measure?). Can we get an intuition for no change in Y-maze tests and what that means biologically?

The authors need to comment on the fact that they often did not see correlations between daily average running and blood flow changes, IOSI (Figure 5), and density (Figure 6) but do see differences in the novel object but not the Y-maze behaviors.

---

## [Author Response]

Essential revisions:1) The use of female mice only should be justified in the manuscript. The authors should discuss the hormonal effect on blood flow.

Thank you for raising this point. We used female mice because our previous data used for comparison with the present findings also used females [4-5]. In addition, we expected female mice to be more affected by exercise than male mice because they tend to be more active compared to males, as exemplified by running longer distances in a 24h period and showing increased cardiac hypertrophy due to running [1-3].

We agree with the Reviewers’ concerns that the observed effects may vary depending on the sex of the animal and should be both discussed in the manuscript and addressed in future studies. We have added the following paragraph to the Discussion and cited relevant literature in the revised manuscript [6-7].

(Lines 373-379, Discussion)

“The present study used female mice because the comparison with our previous studies in young adult female mice and prior data have shown that in comparison to male mice, female mice exhibit higher voluntary running activity that could potentially produce a greater effect on the brain.(Jones et al., 1990; Konhilas et al., 2015; Rosenfeld, 2017) However, the effect may vary in male mice as the animal sex could differentially affect the age-related and exercise-induced changes in cerebral microcirculation due to sex-related hormonal differences.(Bisset et al., 2022; McMullan et al., 2016)”

2) Figure 3: see comments 3 to 5 by reviewer 2 and comments 1 to 10 by reviewer 3. All the points need to be clearly addressed and the manuscript modified accordingly.

Thank you for the comments. We have provided responses to all the points raised by the Reviewers and revised the manuscript accordingly.

3) Figure 5: the authors explain how whisker stimulation led to a decrease in the baseline of the signal (rev.2, comment 6).

We agree that having Figures 1a and 1b combined into one figure will help readers to better compare RBC flux in young versus aged mice. In young sedentary mice, the gray matter RBC flux was measured only between 300-600 µm*,* which corresponds to the cortical layer IV. In aged mice, more imaging planes were added to cover a cortical depth range from layer II/III to IV. This created an inconsistency between young and aged mice in the depth location where the RBC flux was measured. To better compare the capillary RBC flux in young and aged mice, we prepared additional 6 young sedentary mice and performed flux measurements following the same protocol used for aged mice. The result is presented in Figure 1a, showing capillary RBC flux across different brain regions in each animal group, including the young sedentary group.

4) Figure 6: The authors should show the same vascular density analysis in the young cohort. See detailed comment # 11 by reviewer 3.

Thank you for pointing this out. We found that there was a mistake in plotting the regression lines in Figure 1d for both aged sedentary and exercise group. We apologize for the confusion. The correction has been made to Figure 1d in the revised manuscript.

5) The authors need to address the fact that these density measurements are done in superficial cortical areas where they do not see pronounced age-related changes in RBC flux. What might the density of deeper regions be changing with aging and what effect would exercise have (comment 12 / rev. 3)?

The measured flow values from all the venules were first averaged to obtain the mean flow for each mouse. The mean flow values for each group were then averaged over mice from that group (e.g., the STD was calculated based on the number of animals). We now provide more details about the plots in the figure caption. We agree that the larger standard deviation in the exercise group compared with sedentary controls may partly be due to a smaller sample size. The variation may also be related to the variation of the vessel diameter (please see revised Figure 3d). However, no differences in the mean and standard deviation of vessel diameter between the two groups were found (aged sedentary group: 25.1±3.6 µm vs aged exercise group: 27.6±3.6 µm, mean± SD). We have revised our manuscript as follows:

(Lines 116-120, Results)

“Aged exercise group showed a larger variation of the venular flow than aged sedentary group, possibly because of the smaller sample size compared with aged sedentary group. The variation may also be related to the variation of the vessel diameter (Figure 1d) although no significant differences in the mean and standard deviation of vessel diameter between two groups were found (aged sedentary group: 25.1±3.6 µm vs aged exercise group: 27.6±3.6 µm, mean± SD)”

(Figure caption for Figure 1e)

“(e) Mean venular flow in ascending venules in (d) in aged sedentary and exercise group. The measured flow values from all the venules were first averaged to obtain the mean flow for each mouse. The mean flow values for each animal group were then obtained by averaging over mice from that group.”

6) The authors should include a reference to Cudmore et al. (2017) where they perform awake longitudinal in vivo 2P imaging to determine if aerobic exercise can induce changes in capillary structure (length, tortuosity, diameter), angiogenesis, or rarefication. See details in comment 13 by reviewer 3.

We agree that the paper from Cudmore serves as a good reference for our study. The following revision has been made accordingly.

(Lines 322-328, Discussion)

“However, other studies showed no improvements in cerebral microvascular structure with regular exercise in the mouse sensorimotor cortex.(Cudmore et al., 2017; Dorr et al., 2017; Falkenhain et al., 2020) Exercise can affect the brain through different mechanisms across different brain regions at various points in the lifespan.(Stillman et al., 2020) Although it is unclear whether the difference in age or brain region (or both) contributed to the inconsistent results with some of the previous observations, we found significant morphological changes in the microvascular morphometric parameters due to chronic exercise in the somatosensory cortex of 20-month-old mice (Figure 4).”

7) The authors need to comment on the fact that they often did not see correlations between daily average running and blood flow changes, IOSI (Figure 5), and density (Figure 6) but do see differences in the novel object but not the Y-maze behaviors.

We agree and hope that this response can also address the Comment 7 from Reviewer 2.

The following revision has been made in the revised manuscript.

(Lines 366-370, Discussion)

“Among different optical measurements performed in this study, only the capillary RBC flux in subcortical white matter tended to correlate with the running activity while no correlations were found between the parameters measured in cortical gray matter (e.g., venous flow, capillary pO_!_, cortical hemodynamic response, and capillary density) and the average running distance. This could also be attributed to higher responsiveness of subcortical white matter to exercise compared with the cortical gray matter.”

(Lines 347-354, Discussion)

“Consistent with our findings, microcirculatory changes in the white matter have been associated with recognition memory function evaluated by NORT in rodents, while no associations were found with spatial memory function assessed with Y-maze test in the same studies. (Blasi et al., 2014; Choi et al., 2015) However, we do not rule out the possibility that exercise may improve spatial memory function and suggest further investigation to evaluate different aspects of spatial memory function by applying other behavioral tests (e.g., Morris water maze test). As previously shown in several studies in mouse models of AD, each memory task depends on a variety of brain regions that can be differentially affected by exercise, which could also produce inconsistent results between different memory tasks. (Kraeuter et al., 2019; Winters, 2004).”

8) Scatter plots showing individual data points are encouraged

The revision has been made as suggested. Please see revised Figures 1-5, Figure 1 —figure supplements 2-3, and Figure 3 —figure supplement 1.

Reviewer #1 (Recommendations for the authors):1. The authors used female mice only without explaining why. Indeed, using female mice is very important as too many studies use only male mice and thus neglect to study females. Nevertheless, ideally, using both male and female mice would be better. As female mice do not have menopause but have a progressive decline in circulating estrogens the authors could discuss the possible link between estrogen and cerebral blood flow in their study. Another way to investigate the role of estrogens would be to use intact and ovariectomized female mice.

Thank you for raising this point. Please see our response to Essential revision 1 above.

2. As mice were housed individually one may suspect a change in cerebral and cognitive function due to the stress of isolation as mice are very social animals. Although all the groups were submitted to the same experimental conditions, the authors could add a statement about this possible limitation in the discussion.

Thank you for the comment. We have added the following paragraph in the Discussion and cited relevant papers.

(Lines 379-384, Discussion)

“Although all the groups were subjected to the same housing conditions, isolated housing could be considered a stressor that can influence the observed changes in the cerebral microcirculation and behavioral function. Aerobic exercise has been shown to enhance cell proliferation in the dentate gyrus regardless of housing conditions in young and aged mice. (Kannangara et al., 2011) Further investigation into the influence of environmental enrichment factors on the age-related and exercise-induced changes will help to identify the mechanisms that underlie these processes.”

3. Whereas the effect of aerobic exercise is clear in old mice it would be interesting to know if the effect of exercise is equivalent in young mice. In other words, is the flow and pO2 gain equivalent in young and old mice with exercise? Same question with the other parameters.

We assume that aged mice or diseased mice deficient in exercise-inducible biomolecules could be more sensitive to exercise compared with younger healthy controls as exercise may affect the brain through different mechanisms at various points in the lifespan [8]. While better understanding the effect of exercise on the brain microcirculation in the young mice is certainly important, we believe that this question is outside the scope of this work, and it will be addressed in the future studies.

Reviewer #2 (Recommendations for the authors):This manuscript is well written and the evidence presented is convincing that aerobic exercise helps normalize cerebrovascular function. The effects were more evident in deeper cortical tissues and in the white matter. I have a number of comments/questions for the authors to consider:1. Why only female mice? age-dependent changes in female mice could be crucial for the observed effects which might vary in male mice. It would be useful to know whether male mice display a similar pattern.

Thank you for raising this point. Please see our response to Essential revision 1 above.

2. In the exercise group, mice had 5 months of access to exercise wheels. Were the running wheels removed from the cages between months 19-21?

We thank the reviewer for this question. Mice in the exercise group were continuously housed in cages with running wheels during both training and imaging weeks (i.e., between months 19-21). Optical measurements were performed during 5-6 weeks in each animal, and 7-10 days of a break were given between measurements.

(Lines 95-96, Methods)

“Mice in the aged exercise group were continuously housed in cages with running wheels during both training and imaging weeks.”

3. The data presentation in Figure 3a and 3b doesn't clearly compare RBC flux in young versus aged mice. This is a key observation in the study and should be highlighted.

Thank you for the comment. Please see our response to Essential revision 2: Comment 3 from Reviewer 2, above.

4. Figure 3e: It is not clear how the authors obtained the best-fit result for the exercise group (dotted line), since scattered points seem to follow.

Please see our response to Essential revision 2.

5. Figure 3f: The legend mentions that the data are from 14 and 7 venules from 9 and 6 mice, respectively. What is exactly shown in Figure 3f? Was the standard deviation calculation based on the number of vessels or animals? There are twice the number of recordings from the sedentary group compared to the exercise, which possibly explains in part the very large standard deviation in 3f.

Please see our response to Essential revision 2.

6. I suggest that the authors explain how whisker stimulation led to a decrease in the baseline of the signal. It might not be clear and I cannot find a clear explanation of the calculation of light intensity (Figure 5).

Thank you for the suggestion. Please see our response to Essential revision 3.

7. Since only NORT behavioral test showed significant differences between exercise and sedentary groups, I believe that the evidence is too preliminary at this stage and would need more corroborations.

Please see our response to Essential revision 7.

8. Scatter plots showing individual data points are encouraged.

Please see our response to Essential revision 8.

Reviewer #3 (Recommendations for the authors):Their conclusions are in line with the experimental data provided. I am asking for the following revision below.1. Figure 3 a/b. Could panels a and b be combined? The way it is requires comparing the same y-axis across two plots. Shouldn't they be one plot? If they cannot be one plot, why not?

Yes, in the revised manuscript, we combined them into one plot. Thank you for providing this suggestion. Please find our response above to Comment 3 from Reviewer 2.

2. Figure 3 a/b shows "RBC Flux (RBC/sec)" but the supplemental figure shows "RBC Speed (mm/sec)". Are these the same measurement? If so, can the units be the same? If they are not the same measurement, the authors should explain the differences.

We apologize for the confusion. The RBC flux is defined as the number of RBCs that pass through the vessel per unit time (RBC/s), while the RBC speed represents the average speed of the RBCs inside the vessel (mm/s). They were obtained from the same point measurements of the RBC passages through the excitation focal volume. While our measurements provide a direct assessment of the RBC flux, the RBC speed was derived from these measurements by assuming the average RBC length in the capillary. We have provided further details on how we defined these values in the Methods Section in the revised manuscript.

(Lines 532-535, Methods)

“We assessed the RBC flux, RBC speed, and RBC line-density within individual capillaries, extracted from the point-based acquisition of fluorescence signal. The capillary RBC flux was defined as the number of RBCs that passed through the vessel per unit time (RBC/s).”

(Lines 537-542, Methods)

“The capillary RBC speed was defined as the average speed of the RBCs in the vessel (mm/s). Following our previously described procedures,(Li et al., 2019) RBC speed for each RBC-passage event (valley) in the segmented curve was estimated as v=D⁄∆t, where D is RBC diameter, assumed to be 6 μm, and ∆t is the width of the valley. For each capillary, the speed values estimated from each of the valleys in the segmented curve were averaged to obtain the mean RBC speed.”

3. Is this difference because RBC Flux (RBC/sec) is measured with a parked layer and peak/throughs are counted versus RBC Speed (mm/sec) is measured by a more common line scan along the capillary? If this is the case, it needs to be explained.

Please see our response above to Comment 2 from Reviewer 3. Both RBC flux and RBC speed were obtained from the same point-scan measurements of the RBC passages through the excitation focal volume.

4. Figure 3 – Supplemental 2. It is not clear to me what the "RBC line-density" tells us. I understand the definition, "RBC line-density was calculated as the ratio between combined time duration of all valleys to the total duration of the entire time course". The reader needs to be told how to interpret larger or smaller RBC line-density. Please describe this in more detail and link it to a biological intuition or hypothesis.

Thank you for this comment. The capillary RBC line-density measures the fraction of the capillary length occupied by RBCs, which is in general positively correlated with blood hematocrit level (Hct). The white matter RBC flux increase in the aged exercise group could be attributed to an increase in either the RBC speed, RBC-line density, or both. We observed a trend of moderate increases both in the RBC speed and RBC line-density in the exercise group compared with sedentary controls (Figure 1—figure supplement 2 and 3), but no statistically significant differences were found between the two groups. This leaves the possibility that multiple factors are involved in the exercise-induced increase in RBC flux in the white matter. We have revised the manuscript as follows:

(Lines 545-547, Methods)

“The capillary RBC line-density measures the fraction of the capillary length occupied by RBCs, which is closely related to the capillary hematocrit. The capillary RBC line-density was calculated as the ratio between the combined time duration of all signal valleys to the total duration of the entire time course.”

5. Figure 3 d. "Exercise decreased the coefficient of variation (CV) of capillary RBC flux …". Please specify how the std/mean (e.g., CV) was calculated. Was this within a single capillary flow measurement using your parked beam to count the passage of RBCs? Or is it the std/mean across measurements within a region like "layer IV" and a grouping like young or old?

We have added the following paragraph to the Methods Section to specify how the CV of capillary RBC flux was calculated.

(Lines 545-547, Methods)

“The variability in the RBC flux among capillaries, which is termed capillary flow heterogeneity, was quantified by calculating the CV of capillary RBC flux in cortical layers II/III-IV and subcortical white matter. Here, CV is defined as the ratio of the STD to the mean of capillary RBC flux.”

6. Furthermore, I would like the authors to lay out a hypothesis or intuition regarding potential differences in the CV of RBC flux. What does this CV metric tell us? What does it tell us if it is larger and what does it tell us if it is smaller? They show that exercise decreases the CV of RBC flux in white matter but not in the superficial cortex. How do I interpret the meaning of this, what should I take away from this?

The decreased capillary RBC flux heterogeneity (e.g., decreased CV of the capillary RBC flux) is linked with the increased efficiency of oxygen delivery to the tissue [9-10]. In this study, exercise was associated with increased mean capillary RBC flux and decreased CV of the RBC flux in subcortical white matter, suggesting larger and more homogeneous microvascular blood flow in that region that could potentially facilitate improved oxygen delivery to tissue. We expanded the Discussion section with the following sentence:

(Lines 253-256, Discussion)

“Exercise was associated with a decreased CV of the capillary RBC flux in subcortical white matter, suggesting more homogeneous microvascular blood flow in that region that could potentially facilitate improved oxygen delivery to tissue.”

7. Figure 3 – Supplemental 3. The authors look for correlations between average daily running and the RBC flux. I would like to see this analysis using the total distance run per individual. This could be mentioned in the Results section. This kind of total running distance per individual (versus daily averages) could also be used for analysis in other sections such as "3.4 Cortical microvascular density is significantly larger in the exercise group" and behavior experiments in Figure 7b-e.

Thank you for your suggestion. The figure below shows correlations between the total running distance and the RBC flux. No correlations with other measured parameters (e.g., venous flow, intravascular pO_!_, hematocrit, microvascular density, and hemodynamic response amplitude) were found and thus, not shown here. Because of approximately the same number of days that all mice were running, total and average running distances share a common scaling factor (e.g., number of days) and show nearly identical pairwise correlations with the various measured parameters. Therefore, we believe that providing additional figures in the manuscript with the total running distances may not be very informative. We expanded the Discussion section to clarify this connection between total and average running distances in our data.

**Author response image 1. sa2fig1:** Correlations between capillary flow and running activity in the aged exercise group. (a) and (b) Correlations of the average running distance with the white matter capillary RBC flux and CV, respectively. (c) and (d) Correlations of the total running distance with the white matter capillary RBC flux and CV, respectively. (a) and (b) Correlations of the average running distance with the gray matter capillary RBC flux and CV, respectively. (c) and (d) Correlations of the total running distance with the gray matter capillary RBC flux and CV, respectively.

**Author response image 2. sa2fig2:** Correlations between behavioral performance and running activity in the exercise group. (a-d) Correlations between the total running distance and four different DI scores: 1, 2, 3, and 5 minutes, respectively. (a-d) Correlations between the average running distance and four different DI scores: 1, 2, 3, and 5 minutes, respectively.

(Lines 370-373, Discussion)

“Correlations between the measured parameters and the total running distance were analyzed. Because of approximately the same number of days that all mice were running, total and average running distances share a common scaling factor (e.g., number of days) and show nearly identical correlations with all of the parameters measured (data not shown).”

8. Figure 3 f. Venous flow is measured with OCT. Why did the authors not measure arteriole flow in the same manner? For example, arterioles were measured for PO2 in figure 4. One hypothesis is that if exercise increased arteriole flow in the aged group, that would partially explain an increase in venous flow. One confounding parameter here is heart health. If the aged exercise group, simply had stronger flow coming from the heart.

The health of the heart is indeed one of the likely reasons for the increased venous flow observed in the aged exercise group, in addition to potential group differences in the blood flow regulation in the brain, and vascular anatomical differences. On the other hand, observed depth dependent differences in the microcirculation may be more difficult to link with the heart output.

Unfortunately, flow measurement in arterioles could not be performed with our OCT setup because the flow in many diving arterioles exceeded the maximum measurable limit of the flow measurement by Doppler OCT. Details are explained in the Methods Section in the revised manuscript.

(Lines 505-508, Methods)

“For OCT Doppler measurements, the maximum measurable flow speed without phase wrapping was ±12 mm/s when the direction of flow was parallel to the OCT beam and the minimum detectable velocity was ±0.7 mm/s which was determined by the phase noise of the system, measured as ±0.2 radians.”

(Lines 586-588, Methods)

“Flow data measured in diving arterioles were excluded from analysis due to their much higher flow compared to venular flow that often causes excessive phase wrapping and signal fading. (Koch et al., 2009)”

We expanded the Discussion section with the following text:

(Lines 264-268, Discussion)

“The increased venous flow in the exercise group could be associated with an increased arterial flow, possibly affected by exercise-induced changes in heart strength, brain blood flow regulation, and vascular anatomy. However, flow measurement in arterioles could not be performed with our OCT setup because the flow in many diving arterioles exceeded the maximum measurable limit of the flow measurement by Doppler OCT.”

9. Figure 3 c, 4 b. I would like the authors to consider showing these histograms as cumulative histograms. It may be easier for the reader to visually see the shift in the mean of the distribution (compared to a histogram).

The previous histograms (Figure 1c and 2b in the original manuscript) were included in the supplementary data (Figure 2 —figure supplement 1).

(Lines 97-98, Results)

“Cumulative histograms of capillary RBC flux in gray and whiter matter confirmed this finding (Figure 1b and Figure 1—figure supplement 1).”

10. "3.3 Cortical hemodynamic response to functional activation was reduced by aging but not altered by exercise" and "Figure 5". The functional hemodynamics measured with OISI is using a video camera focused on the cortical surface and thus represents sensory stimulus-invoked changes in blood flow in superficial cortical areas. Throughout the manuscript, the findings reveal there are few exercise-induced changes in these superficial cortical areas. The authors need to address this in the text. The fact there is no effect of exercise on the superficial evoked blood flow is not at all surprising given many of the other results.

Thank you for the comments. The revision has been made as follows:

(Lines 305-309, Discussion)

“Consistent with the small effect of exercise on cerebral microcirculation and oxygenation in superficial cortical areas, we observed no change in the relative peak amplitude or the latency of stimulus-induced hemodynamic response with exercise, assessed with OISI at 570 nm, which emphasizes the intrinsic signal originating from cerebral blood volume changes in the superficial cortical layers (Figure 3 and Figure 3 —figure supplement 1).”

11. Figure 6. Can the authors show the same vascular density analysis in the young cohort? The hypothesis is that aging reduces vessel density. This would be verification in that it would replicate what has previously been reported and could act as a baseline to compare and discuss the changes in density seen in aged sedentary versus exercising groups. How far does the increase in density in aged exercising mice bring the capillary density toward the young sedentary condition?

The LED used for OISI had a center wavelength of 570 nm, which is close to the hemoglobin isosbestic point. Therefore, the optical intrinsic signal at this wavelength is primarily sensitive to the total hemoglobin concentration, and thus blood volume changes under the assumption that hematocrit is not changing. Increases in local cerebral blood volume during functional hyperemia in response to whisker stimulation correlate with decreased diffused reflectance i.e., decreased optical intrinsic signal. To avoid confusion, a traditional way of presenting this data is to use the absolute value of relative signal change with respect to the baseline (e.g., to invert the signal) so that the increase in the blood volume is presented as an increase in the signal. Figures 3a and 3b have been revised accordingly.

12. The authors need to address the fact that these density measurements are done in superficial cortical areas where they do not see pronounced age-related changes in RBC flux. What might the density of deeper regions be changing with aging and what effect would exercise have?

Please see our response to the Essential revision 5.

13. I want the authors to consider including a reference to Cudmore et al. (2017) where they perform awake longitudinal in vivo 2P imaging to determine if aerobic exercise can induce changes in capillary structure (length, tortuosity, diameter), angiogenesis, or rarefication. This study by Cudmore et al. was performed in young adult mice and they conclude aerobic exercise (voluntary wheel running) does not alter capillary density. It would be interesting to comment on their results (no effect) and your results in aged and exercising mice (big effect).Cudmore RH, et al. (2017) Cerebral vascular structure in the motor cortex of adult mice is stable and is not altered by voluntary exercise. J Cereb Blood Flow Metab, 37(12):3725-3743. doi: 10.1177/0271678X16682508.

Please see our response to the Essential revision 6.

14. Figure 7. In the behavioral experiments it was found that exercise changed novel exploration but not the Y-maze test. The authors need to comment on why they think this is the case. I guess exercise changes short-term memory (novel object) versus Y-maze (what does the Y-maze measure?). Can we get an intuition for no change in Y-maze tests and what that means biologically?

Thank you for this comment. We have revised the manuscript as follows.

(Lines 347-354, Discussion)

“Consistent with our findings, microcirculatory changes in the white matter have been associated with recognition memory function evaluated by NORT in rodents, while no associations were found with spatial memory function assessed with Y-maze test in the same studies. (Blasi et al., 2014; Choi et al., 2015) However, we do not rule out the possibility that exercise may improve spatial memory function and suggest further investigation to evaluate different aspects of spatial memory function by applying other behavioral tests (e.g., Morris water maze test). As previously shown in several studies in mouse models of AD, each memory task depends on a variety of brain regions that can be differentially affected by exercise, which could also produce inconsistent results between different memory tasks. (Kraeuter et al., 2019; Winters, 2004).”

The authors need to comment on the fact that they often did not see correlations between daily average running and blood flow changes, IOSI (Figure 5), and density (Figure 6) but do see differences in the novel object but not the Y-maze behaviors.

Please see our response to the Essential revision 7.